



# Using an oceanographic model to investigate the mystery of the missing puerulus

Jessica Kolbusz[1], Tim Langlois[2], Charitha Pattiaratchi[1,] Simon de Lestang[3]

[1] Oceans Graduate School and the UWA Oceans Institute, The University of Western Australia, Crawley, WA 6009, Australia
[2] School of Biological Sciences and the UWA Oceans Institute, The University of Western Australia, Crawley, WA, Australia
[3] Western Australian Fisheries and Marine Research Laboratories, Department of Primary Industries and Regional Development, Government of Western Australia, North Beach, WA, Australia

*Correspondence to*: Jessica Kolbusz (jessica.kolbusz@research.uwa.edu.au)

**Abstract.** Dynamics of ocean boundary currents and associated shelf processes can influence onshore/offshore transport of water, critically impacting marine organisms that release long-lived pelagic larvae into the water column. The western rock lobster, *Panulirus cygnus*, endemic to Western Australia, is the basis of Australia's most valuable wild-caught commercial fishery. After hatching, western rock lobster larvae (phyllosoma) spend up to 11 months in offshore waters before ocean currents and their ability to swim, transport them back to the coast. The abundance of western rock lobster puerulus (settlement
phase post phyllosoma) has historically been observed to be positively correlated with the strength of the Leeuwin Current, and an index of puerulus numbers is used by fisheries managers as a predictor of subsequent lobster abundance 3-4 years later. In 2008 and 2009 the Leeuwin Current was strong, yet a settlement failure occurred throughout the fishery prompting management changes and a rethinking of environmental factors associated with their settlement. Thus, understanding factors that may have been responsible for the settlement failure is important for fisheries management. Oceanographic parameters
likely to influence puerulus settlement were derived for a 17 year period to investigate correlations. Analysis indicated that puerulus settlement at adjacent monitoring sites have similar oceanographic forcing with kinetic energy in the offshore and the strength of the Leeuwin Current being key factors. Settlement failure years were synonymous with 'hiatus' conditions in the south-east Indian Ocean, and periods of sustained cooler water present offshore. Post 2009, there has been an unusual but consistent increase in the Leeuwin Current during the early summer months with a matching decrease in the Capes Current,
that may explain an observed settlement timing mismatch compared to historical data. Our study has revealed that a culmination of these conditions likely led to the recruitment failure and subsequent changes in puerulus settlement patterns.

## 1 Introduction

Fisheries management of the western rock lobster (*Panulirus cygnus*), Australia's most valuable wild-caught single-species fishery (de Lestang et al., 2018), utilises an index of *P. cygnus* post-larvae (puerulus) settlement as one of its main
stock diagnostics. Over the past four decades, this index has been used to predict catches 3 to 4 years in advance (Phillips,





1986; Caputi and Brown, 1993; de Lestang et al., 2015) and puerulus settlement was historically observed to be positively correlated with the strength of the Leeuwin Current (Pearce and Phillips, 1988; Lenanton et al., 1991). During the 2008 and 2009 settlement seasons (May - April) there was an unexpected, given the strong Leeuwin Current over those years, settlement failure. In response to this, the Department of Primary Industries and Regional Development, Western Australia (DPIRD, WA)

fisheries managers made large reductions to landings and restructured the management system from input to output controls. Puerulus settlement has subsequently recovered, but despite extensive research, no clear factor(s) explaining the settlement failure have been identified to date. Recent research has shown that, since the recovery in puerulus numbers, there has been a latitudinal and timing shift in settlement compared to historical data (Kolbusz et al., 2021). This study aims to better understand why this recruitment failure occurred in 2008 and 2009, and why there has been a change in the timing and latitude before and

after these years, particularly in relation to oceanographic processes.

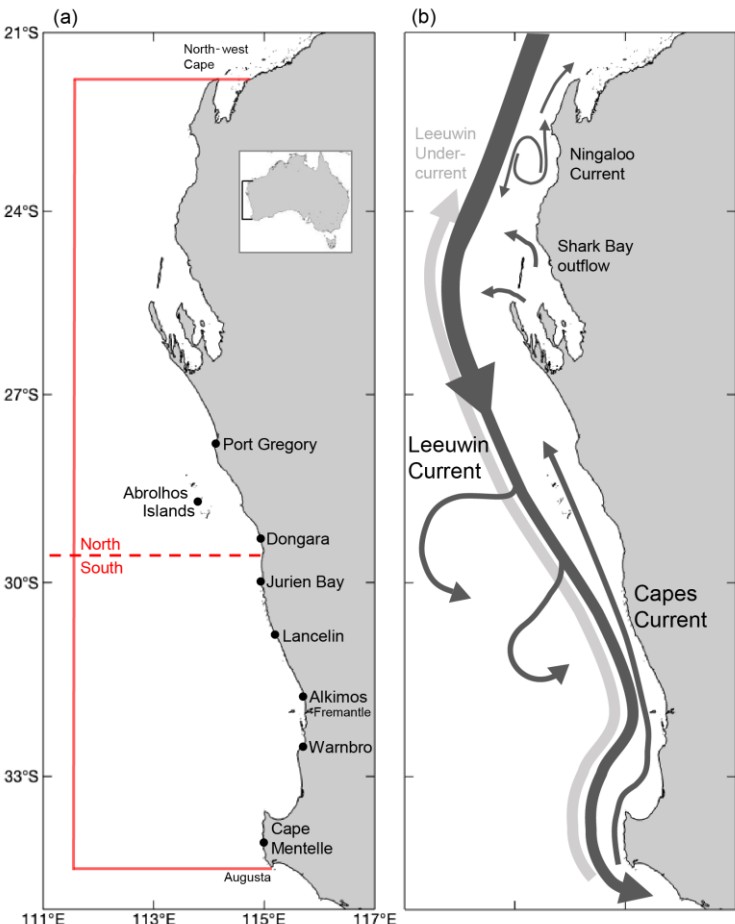

**Figure 1 (a) Locations of puerulus survey sites included within this study and other locations of note. The north and south refer to the midpoint used in this analysis. In particular for kinetic and eddy kinetic energy calculation. Red boundary is the extent of environmental variables. Inset shows location of the coastline in Australia (b) Schematic of the major currents systems thought to**
**influence early-stage *P. cygnus* larvae movement. Relative arrow size and location show characteristic currents. Eddies generated by the LC flow down the continental shelf are seen.**



Between November and February, berried western rock lobster females release their larvae (phyllosoma) throughout the study region (Figure 1a), which are then transported offshore by the prevailing ocean currents into deeper ocean where they transform over nine larval stages through a series of temperature-dependent moults (Figure 2). At around eight months, they undergo their final metamorphosis into the actively swimming nektonic puerulus (post-larval stage) (Figure 2). The onshore transport and movement across the continental shelf occurs mainly during August – January (late austral winter-summer to settle in shallow areas of generally less than 5 metres depth (Phillips, 1981; de Lestang et al., 2018). Circulation patterns of the south-east Indian Ocean likely influence spatially varied cross-shelf transport of the puerulus (Caputi, 2008; Feng et al., 2011a).

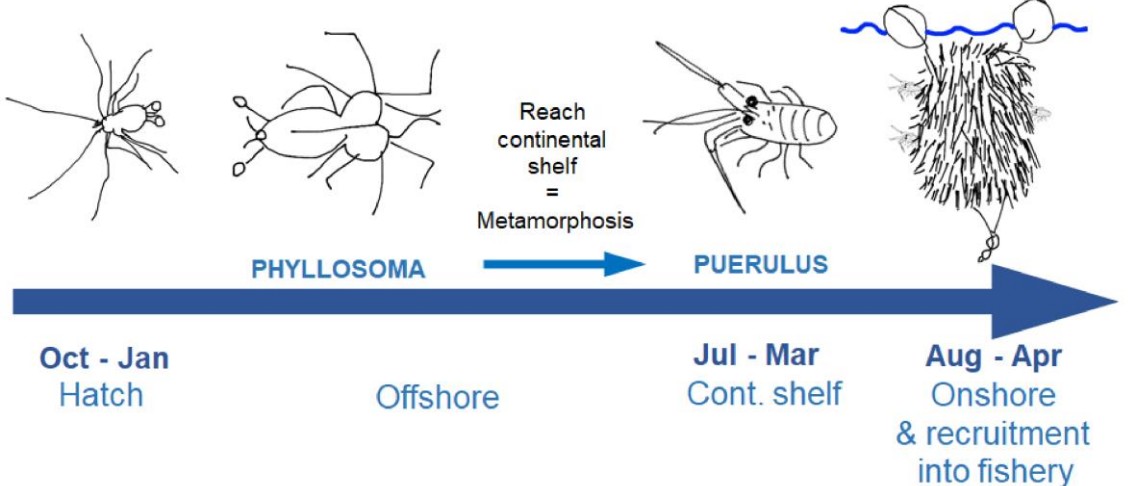

**Figure 2 Early life-cycle schematic of *P. cygnus*. Below the arrow indicates the approximate timing and location of each stage. Above the arrow displays their growth and eventual recruitment into the fishery where they are 'collected'.**

In the late 1960's puerulus collectors resembling artificial seaweed were developed and deployed at several shallow-water sites within the fishery (Phillips 1981). Puerulus numbers are currently collected at eight sites spanning the latitude of the fishery with the centrally-located Dongara collectors (Figure 1) now providing over 50 years of *in-situ* data (Kolbusz et al., 2021). Since majority of the settlement occurs between August and January, and very little occurs between April and May, the puerulus settlement index derived from these data (puerulus index, PI) is standardised to a 12-month period spanning May to the following April and is the mean of the sum of puerulus counted at all sites during each full moon (i.e. approximately monthly).

Research on the interaction between the physical environment and PI began in the 1980s with a strong positive relationship found between the strength of the Leeuwin Current (LC), with Fremantle Mean Sea Level (FMSL) as a proxy (Pearce and Phillips, 1988; Lenanton et al., 1991). Consequently, La Niña phases were also found to coincide with an above-average PI, thought to be due to a strengthened LC during these phases, with the Southern Oscillation Index (SOI) as an indicator of El Niño Southern Oscillation phases (Pearce and Phillips, 1988; Clarke and Li, 2004; Caputi et al., 2001). These





relationships were identified through long-term correlations between the PI and both FMSL and SOI (Figure 3). Since 1988, studies have also demonstrated the inter-annual variation in PI was influenced by the sea surface temperature (SST) and westerly (onshore) winds (Caputi and Brown, 1993; Caputi, 2008). Caputi et al. (2001) defined a significant area overlapping with the spatial extent of the LC, where SST (27°–34°S, 105°–117°E) in February-April had a positive relationship with the PI of the subsequent season. Bottom temperature during spawning has also been identified as a cue for spawning and possible

influence on the puerulus settlement season to follow (Chittleborough, 1975; de Lestang et al., 2015). In addition, westerly winds, associated with storms in late winter to spring, have been related to an increased PI for the coming season (Caputi et al., 2010).



**Figure 3 Time series of annual (a) fishery-wide Puerulus Index (May – April); (b) Fremantle Mean Sea Level (FMSL) over June to**
**December (m); and, (c) Southern Oscillation Index (SOI) (May –April). Grey shaded seasons indicate the less than expected PI based on a priori relationships (Caputi et al. 2001). Blue and red shading indicates a La Niña and El Niño periods, respectively. Updated to 2018 and modified from Figure 5 in Caputi et al. 2001.**

Over two consecutive years (2008 to 2009), an abnormally low PI occurred prompting management changes and a re-evaluation of environmental drivers of puerulus settlement (Figure 3) (Caputi et al., 2014; de Lestang et al., 2015) (de

Lestang et al. 2014; Caputi et al. 2014). In 2008, the settlement was expected to be above average due to the presence of warm



offshore water temperatures in early 2007 and a high FMSL (Figure 3), but instead was a record low, which was then followed by the second lowest settlement in 2009.  Below average PI was recorded for seven consecutive seasons (2006 – 2012) (Feng et al., 2011b; de Lestang et al., 2015) and through one of the strongest La Niña phases on record (2011) (Boening et al., 2012). Following this period of decline, previous relationships between PI and oceanographic factors have broken down and the

correspondence between FMSL and SOI and puerulus settlement no longer exist (Figure 3) (Feng et al., 2011a; de Lestang et al., 2015; Caputi et al., 2014). These changes have been attributed to shifting oceanographic conditions relating to increasing water temperatures during the spawning period resulting in an earlier onset of spawning and a decrease in the number of storms occurring near the time of puerulus settlement (de Lestang et al., 2015).

Since 2009, the majority of the reduction in the settlement of puerulus has occurred in the first half of the settlement

season (May - October; Kolbusz et al., 2021). Additionally, in the southern sites, there has been a significant reduction over the whole season, whereas those in the north have maintained levels of settlement during the second half of the season. Prior to current study, research on the 2008/09 decline had included an examination of overfishing of the spawning stock (de Lestang et al., 2015) and whether conditions of survival were no longer met in the early pelagic life stages (Säwström et al., 2014). Given the recent research revealing a mismatch in the timing of settlement, compared to historical norms (Kolbusz et al., 2021)

the current study set out to further investigate the environmental factors that may have led to the changes in puerulus settlement.

Since the apparent decoupling of some oceanographic parameters and puerulus settlement and the initial research into the anomaly, additional years of contrasting settlement have been recorded, thus providing a larger dataset to re-examine these relationships following the period of low settlement. In addition, recent studies have shown changes in inter-annual ocean variability in both the southwest of Australia and along the whole coastline, with an increase in the number of storms occurring

and shifting to later in the year (Wandres et al., 2017). Studying direct physical oceanographic parameters likely to influence phyllosoma on their offshore and eventual coastal life cycle, provides an opportunity to investigate local variation in puerulus settlement or whether long term trends and lagging effects are at play For example, the causal factors of the correlation between LC strength and puerulus settlement index prior to 2009 is unknown, whether it was due to the warmer waters the LC brings, or higher eddy retention of larvae close to the coast and therefore better nutritional development of *P. cygnus* (Caputi et al.,

2001; O'Rorke et al., 2014; Wang et al., 2015; Lenanton et al., 1991). On the shelf, the northward flowing Capes Current (CC) is suspected to correspond with the offshore movement in the early larval stages (October – March) from the coast (Feng et al., 2011a).

The recent availability of high resolution numerical oceanographic model output over an extended period (Wijeratne et al., 2018) eliminates the need to use proxies to represent oceanographic parameters likely to influence larvae transport.

Alongside direct predicted values of the LC, CC and temperature we were able to calculate oceanographic parameters which have not been previously examined against puerulus settlement. This includes kinetic energy (KE), eddy kinetic energy (EKE) and cross-shelf transport. The aim of this paper is to use numerical model outputs and satellite data (sea surface temperature, SST) to identify physical oceanographic variables, and biological information on estimates of western rock lobster spawning biomass, to investigate observed variations in PI over the past two decades.



## 2 Study region


Water circulation off the west coast of Australia is driven by the Leeuwin Current (LC) System that incorporates the Leeuwin Undercurrent, and summer wind-driven currents, the Capes (CC) and Ningaloo (NC) currents on the continental shelf (Figure 1) (Woo and Pattiaratchi, 2008; Pattiaratchi and Woo, 2009). The LC is generated through a meridional pressure gradient resulting from the difference between lower density water off northwest Australia and the denser water of the Southern

Ocean (Hamon, 1965; Pearce and Phillips, 1988; Pattiaratchi and Buchan, 1991). The mean southward volume transport of the LC peaks around 32.8º S, through input from South Indian Counter Current (Wijeratne et al., 2018). Near 27 – 28º S statistical analysis has shown that there is a break point in the LC suggesting responses from the current's forcing along the coastline may differ either side of this latitude (Chittleborough, 1976; Berthot et al., 2007). The El Niño Southern Oscillation cycle causes the pressure gradient to decrease/increase during an El Niño/La Niña episode, resulting in a weaker/stronger LC and

cooler/warmer SST (Pattiaratchi and Buchan, 1991; Feng et al., 2003; Wijeratne et al., 2018). This is supported with strong correlations found between the SOI and LC transport at 34º S with a 6-month lag (Schiller et al., 2008).

Surface current variability on the continental shelf within the study region is predominantly wind-driven (Smith et al., 1991; Pattiaratchi et al., 1997). The LC is stronger during austral winter (May – July) and weaker during the austral summer (November- March) due to variations in equatorial wind stress and the Australasian monsoon season (January – March)

(Pattiaratchi and Siji, 2020; Pattiaratchi and Woo, 2009; Smith et al., 1991; Wijeratne et al., 2018) (Smith et al. 1991; Pattiaratchi and Woo, 2009; Wijeratne et al. 2018; Pattiaratchi et al., 2020). A weaker secondary peak in the LC also occurs over December/January (Wijeratne et al., 2018). During the summer months, southerly wind stresses overcome the alongshore pressure gradient, moving upper layers offshore and favouring upwelling onto the continental shelf (Pearce and Pattiaratchi, 1999). The CC can be identified through cooler waters, inshore of the LC and usually inshore of the 50 m contour, initiated

~34º S with the cooler water extending to 27º S (Figure 1) (Gersbach et al., 1999). The LC migrates offshore and is weaker over these predominantly sea-breeze dominated summer months, whereas during autumn/winter it floods the shelf and dominates the distribution of water masses (Cresswell et al., 1989; Pattiaratchi et al., 1997; Woo and Pattiaratchi, 2008).

Mesoscale eddies have been identified in the LC system for more than 30 years (Andrews, 1977; Pearce and Griffiths, 1991; Cosoli et al., 2020). The LC, and associated flows, become unstable with the large variations in topography over the

latitudinal extent of the current, generating eddies, meanders and offshoots (Batteen et al., 2007). The Abrolhos Islands at 28.8º S and the narrowing of the continental shelf slope south of Dongara and the Perth Canyon are major topographic features for the preferential generation of these eddies (Figure 1) (Feng et al., 2005; Meuleners et al., 2008; Rennie et al., 2007; Huang and Feng, 2015; Cosoli et al., 2020). Eddies in LC are mainly generated between 28º S and 33º S (Rennie et al., 2007; Fang and Morrow, 2003; Cosoli et al., 2020). These mesoscale eddies have a mean radius of ~100 km and generally keep their original

formation shape, lasting approximately 8 months (Fang and Morrow, 2003; Cosoli et al., 2020). An increasing strength of the LC is replicated through increases in meanders and eddies down the coast, causing increases/decreases in austral winter/summer and increases/decreases during La Niña/El Niño phases (Feng et al., 2005; Pattiaratchi and Woo, 2009).



## 3 Methods

### 3.1 Puerulus settlement data

Puerulus settlement is surveyed year-round, currently at eight sites across the fishery (between 34 - 27°S) using artificial seagrass-like collectors. Sampling is conducted as close as possible to the full-moon, but may occur five-days either side. Puerulus are likely to have settled on the previous new moon period, giving approximately monthly data (de Lestang et al., 2012). The fishery-wide standardized puerulus index, PI, is calculated based on the seasonal (May – April) mean puerulus settlement numbers from all 8 sites, then summed to obtain an annual index (Kolbusz et al., 2021). The monthly puerulus
settlement at each site is calculated as the average number of puerulus per collector. For this study we used the puerulus settlement data from 2000/01 season to 2016/17 at each of the eight sites (Figure 1), aligned with high resolution oceanographic data (see below) and estimates of western rock lobster spawning biomass. Each site was treated separately, with analysis between seasons being split into "Early" (May – October) and "Late" (November – April) as described by Kolbusz et al. (2021).

### 3.2 Oceanographic data

#### 3.2.1 Numerical model outputs

The Regional Ocean Modelling System (ROMS) has been used in hindcast mode (past-time) for the whole of Australia (ozROMS) to resolve subsurface and surface currents and the associated volume transports (Wijeratne et al., 2018). This is a fully three-dimensional circulation model, resolving processes along the continental shelf, which includes tides, setting it apart from other ocean models for the same region. The grid was set at a horizontal spacing of 3 km to allow for
topographic detail, providing predicted water movement. ozROMS model output is available for the period 2000-2017 for zonal (eastward) and meridional (northward) velocities, as well as temperature. Details of the model are described by Wijeratne et al. (2018) including validation. Examination of this predicted data set allows for the strength of the LC, CC and meridional and zonal transport to be determined, as well as estimates of kinetic (KE) and eddy kinetic energy (EKE) calculations
(Pattiaratchi and Siji, 2020).

Monthly surface KE and EKE was calculated from ozROMS to characterise the variability in the currents. A monthly time series was estimated following Eq. (1) for KE and Eq. (2) for EKE:

$$KE = \sqrt{\frac{u^2 + v^2}{2}}, \tag{1}$$

$$EKE = \sqrt{\frac{u'^2 + v'^2}{2}}, \tag{2}$$

where $u$ and $v$ are the monthly mean meridional and zonal velocities, respectively (Caballero et al., 2008), and $u'$ and $v'$ are the monthly averages with the climatological means subtracted to remove seasonality (Luo et al., 2011). Monthly KE and EKE



values were then averaged to derive an annual mean value to align with the average timing of phyllosoma being offshore (Jan to December, Figure 2). This was divided into a north and south offshore box of the approximate extent where phyllosoma are transported (Figure 1).

The monthly transport estimates of LC and CC (in Sverdrups, Sv = $10^6$ $m^3s^{-1}$) were derived using the ozROMS hindcast dataset (Wijeratne et al., 2018). The transport for each current was defined as follows: (1) CC: northward volume transport of water across latitudes 27°S, 30°S and 34°S in water depths less than 100 m; and, (2) LC: southward volume transport of water across latitudes 27°S, 30°S and 34°S but in water depths greater than 100 m but limited to upper 300 m of the water column (Table A1, Appendix A). For the LC austral winter strength, the transport over June, July and August was

averaged and for summer, December, January and February was averaged. The LC summer period corresponds to recently hatched larvae leaving the continental shelf and puerulus returning in the "Late" portion of the season (Figure 2). The LC winter period corresponds to when puerulus are returning to the shelf. The CC strength was divided into early (September, October and November) and late (December, January, February and March) and corresponded to the time when recently hatched larvae were leaving the continental shelf and puerulus returning in the "Early" and "Late" portions of the season

respectively (Figure 2).

Cross-shelf transport (in Sv) was calculated for each monthly time step between the depth contours (200 – 50 m) over 2 degree latitudinal bins (26 - 28º S, 28 - 30º S, 30 - 32º S and 32 - 34º S) (Table A1, Appendix 1). These latitudinal bins were chosen closest to account for differences in topography and cross-shelf flow differences across the survey sites. Monthly averages for each latitudinal bin indicated that the variability in transport is highest over April to September. Cross-shelf

transport at that time corresponds to the eastward movement of puerulus. Recently hatched larvae cross the shelf (westward) to the open ocean between September and March each season to align with the months that phyllosoma are crossing the shelf (Figure 2). These sets of months were averaged to get two values of cross-shelf transport (westward as phyllosoma and eastward as larvae) for each season (Figure 2).

Temperatures in the model layer immediately above the seabed ('bottom temperature') over the spawning depths (40

– 80 m) were retrieved from ozROMS due to absence of in-situ data. The predicted data were averaged over a northern and southern subset of the data for the whole spawning season (September - March). Due to the ozROMS hindcast starting in 2000, values were only available from 2001. The temperature in the top 100 m of the assumed phyllosoma distribution in south-east Indian Ocean (east of 108ºE) as mean annual value from ozROMS was also included. This accounts for temperature variation over the migrating depths phyllosoma occupy over their pelagic early life-cycle.

**3.2.2 Satellite derived sea surface temperature (SST) and altimetry**

Satellite-derived SST data for the study region were obtained from the Integrated Marine Observing System (IMOS) Australian Ocean Data Network (AODN) portal (portal.aodn.org.au). The climatology data, centred on base period 1993 – 2020, SST Atlas of Australian Regional Seas (SSTAARS) (Wijffels et al., 2018) were used to derive monthly SST anomalies for the region extending offshore to 108ºE (extent in Figure 1, see also (Pattiaratchi and Hetzel, 2020)). This was to reflect the



extensive duration (~9 months) of the pelagic larval and pre-settlement stage *of P. cygnus* (Phillips, 1981). All monthly SST anomalies were originally included in the analysis due to the likely importance of temperature on all stages of the pelagic larval stage (Caputi et al., 2001; de Lestang et al., 2015).

Altimeter data was accessed from the IMOS AODN (portal.aodn.org.au) to further investigate the relationship between the puerulus index and energy in the system. To investigate the long-term KE and EKE over the south-east Indian 220 Ocean this data was applicable (Pattiaratchi and Siji, 2020)

## 3.3 Independent Breeding Stock Survey (IBSS)

Independent Breeding Stock Surveys (IBSS) have been conducted annually since 1992 over the last new moon (~ 15 November) before the start of the historic fishing season (de Lestang et al., 2018). The catch rates of spawning females from this survey (adjusted for fecundity) provide a standardised index of egg production. It is conducted at up to 6 sites spanning 225 the fishery and close to the timing of peak of egg production (November) (Caputi et al., 1995; Chubb, 1991; de Lestang et al., 2016). The IBSS was included in the analysis to account for likely variability in the number of hatching larvae each season.

## 3.4 Multiple regression analysis

The oceanographic variables likely to influence water movement and the distribution and survivorship of *P. cygnus*, detailed above, were considered as predictors of the puerulus settlement within a multiple regression analysis. The IBSS was 230 included as a predictor to include variability in the number of hatching larvae. Due to the large latitudinal range of the settlement data, spatial variability of some environmental variables was incorporated by dividing them into northern, central or southern areas depending on the data type and availability (Appendix 1). A value was obtained for each possible predictor variable to align with each half of the puerulus season.

Generalised additive models (GAMs) with full subset model selection (FSSgam) were used to investigate the 235 influence of 16 different response variables, with the eight sites divided into early and late puerulus settlement periods (Fisher et al., 2018). Models containing variable combinations with correlations > 0.4 were excluded, to eliminate potential problems with collinearity and overfitting (Graham 2003). Due to the relatively small sample size and heterogeneous distribution of the predictors, all were limited to a linear relationship, except cross-shelf transport which was limited to a maximum of three knots per spline, and model sizes were limited to three predictors to prevent overfitting. Model selection was based on Akaike's An 240 Information Criterion (AIC, Akaike, 1973) optimised for small samples sizes (AICc, Hurvich and Tsai, 1989) and the best models were selected at the most parsimonious within two AICc units of the model with the lowest AICc (Burnham and Anderson, 2002). Importance scores for each variable were obtained by summing the AICc weights of each model that each variable occurred within (Fisher et al., 2018).

The R language for statistical computing (R Core Team 2018) was used for all data manipulation (Wickham et al., 245 2018) (dplyr, Wickham et al. 2018) and analysis (Wood, 2017) (mgcv, Wood 2011). MATLAB and the m_map toolbox were used for any spatial plotting (MATLAB, 2019b; M_Map: A mapping package for MATLAB, Version 1.4m).





All variables considered were examined to determine which could be treated as continuous covariates. Linear regression analysis was performed to assess whether there were strong (>0.80) correlations between any variable. Where one was found, a case by case approach was taken to determine whether both, an average, or one of the two variables was to be

250    included in the overall model prior to the time series modelling. Bottom temperatures were all highly correlated (>0.89) and were therefore averaged to give a single variable. Co-correlation between the SST of adjacent months lead to a winter and summer average being used.

The data for each oceanographic and meteorological variable were collated into annual values, and resulted in a total of 39 possible predictors of the puerulus index at sites for late settlement (8 sites), and 33 possible predictors of the puerulus

255    settlement at sites (8) for early settlement (Appendix A). LC strength in summer and late CC strength predictors for early settlement were omitted since they occur after early settlement each season. Considering the large number of predictors and in order to interpret the results, a hypothesis table including each predictor was made (Table 1). Given the predicted data availability and the spawning season being the calendar year prior, the relationship between all predictors and puerulus settlement was limited to the 2001 to 2017 seasons.

260



**Table 1 Predictor and metrics used in multiple regression to investigate variability in puerulus settlement. The subscript *s* identifies the relativity of a month to the puerulus settlement season (May - Apr) in question. *s - 1* is within the season prior and *s + 1* is after.**

| Predictor variable | Metric used | Hypothesised relationship to puerulus settlement |
|---|---|---|
| Leeuwin Current (LC) | Southward strength of the current in Sverdrups (Sv) at northern, central and southern locations for three periods: **1**.Larvae hatching / transport offshore ($Dec_{s-1}$ - $Feb_{s-1}$). **2**. Puerulus transport towards continental shelf ($May_s$ - $June_s$) **3**. Peak settlement ($Dec_s$ - $Feb_s$). | **1**. -ve (all sites). **2**. +ve (all sites). **3**. -ve (north sites), +ve (south sites). |
| Capes Current (CC) | Northward strength of the current in Sv at a north, two central and a south location over two periods: **1**.Larvae hatching / transport offshore ($Sep_{s-1}$ - $Nov_{s-1}$ (early) and $Dec_{s-1}$ - $Feb_{s-1}$(late)) **2**.Peak settlement ($Sep_s$ - $Nov_s$(early) and $Dec_s$ - $Feb_s$(late)) | **1**. +ve (all sites) **2**. +ve (north sites), -ve (south sites) |
| Kinetic and eddy kinetic energy | Kinetic and eddy kinetic energy of the south-east Indian Ocean in $cm^2/s^2$ over a north and south area defined in Figure 1. **1**. phyllosoma offshore ($Jan_{s-1}$ - $Dec_s$) | **1.** +ve (all sites) |
| Cross-shelf transport | Cross-shelf transport over the continental shelf (150 - 50 meters) in Sv over a northern, two central and southern latitudinal bins. **1**.Larvae hatching / phyllosoma transport west ($Sep_{s-1}$ - $Mar_{s-1}$) **2**.Puerulus transport east ($Apr_{s-1}$- $Sep_s$) | **1**. Westerly (-Sv) +'ve **2**. Westerly (+Sv) -'ve |
| Temperature | Water temperature over three periods. **1**. SST summer ($Sep_{s-1}$ - $Mar_{s-1}$) and SST winter ($Apr_{s-1}$ - $Aug_s$) **2**.Bottom temperature during spawning (40 - 60 m depth, $Sep_{s-1}$ - $Mar_{s-1}$) **3**.Top 100 m Early-stage phyllosoma. ($Jan_{s-1}$ - $Dec_s$) | **1**. SST +ve (all sites) **2**. -ve (all sites) **3**. +ve (all sites) |
| Independent Breeding Stock Surveys (IBSS) | **1**. IBSS index for the spawning season | **1.** +ve (all sites) |



## 3.5 Exploration of oceanographic patterns

Subsequent to the results of the multiple regression analysis, findings of importance were expanded upon. Seasonal and inter-annual variability, not captured within the multiple regression analysis, were explored. In particular, the hiatus conditions experienced in the south-east Indian Ocean (Pattiaratchi and Siji, 2020) and interactions between the CC and LC during the summer (Pattiaratchi and Woo, 2009). ENSO information (SOI) and the Fremantle Mean Sea Level (FMSL) was obtained from the Bureau of Meteorology (Fremantle Mean Sea Level; Southern Oscillation Index) (Figure 2). The LC and CC summer period strengths were standardised for each season and then the difference between the two was plotted alongside the early and late puerulus settlement levels.

## 4 Results and Discussion

Results and discussion are combined into three sections (1) time-series patterns of the spatial and temporal variability of the physical environment experienced by *P. cygnus* larvae between 2000 and 2017 (when data was available); (2) exploring correlation of oceanographic conditions with multiple regression analysis, and (3) inter-annual and seasonal oceanographic variability.

## 4.1 Time-series patterns

Puerulus settlement differs dramatically over the latitudes of the fishery with central latitudes experiencing the highest numbers (Figure 4). At the Abrolhos (Figure 4a) the late settlement is consistently higher and remained consistent after the recruitment failure. Other sites, display similar early (Early PI) and late (Late PI) puerulus settlement prior to 2008. However, after 2009, recovery occurs predominately in the later half (Figures 4a, c, d and e).





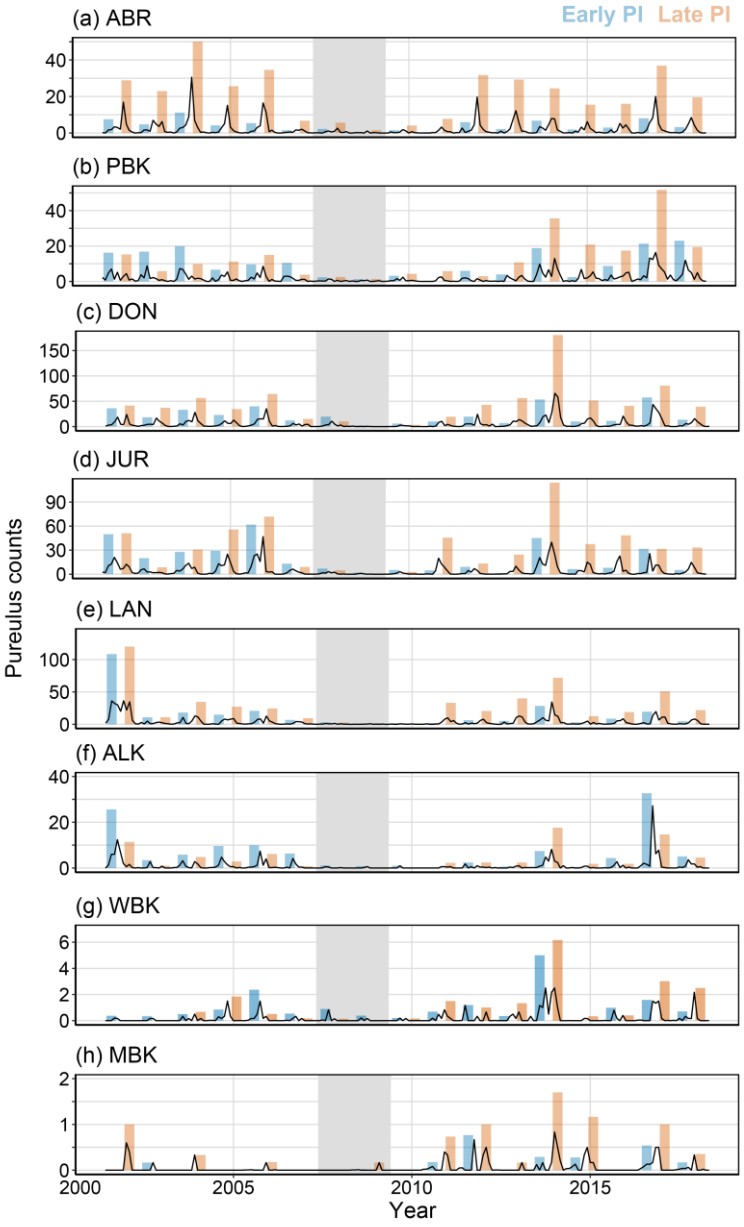

**Figure 4 Monthly average puerulus counts for each monitoring site (black line) with the early (May - October, blue) and late (November - April, red) puerulus index for the season. The index is a sum of the included monthly average puerulus counts. Grey shaded seasons (2008 and 2009) indicate the less than expected PI based on a priori relationships (Caputi et al. 2001).**

The three temperature variables (SST, top 100 m temperature and bottom temperature) all followed a similar pattern to one another (Figures 5 a, b and c). They gradually decreased from highs in 2000 to a low in 2005 before slightly increasing from 2008, all reaching maxima in 2011 when a marine heat-wave occurred in February (Wernberg et al., 2012). A decrease of approximately 1-2 °C in bottom temperature during the spawning season was evident over the early 2000s which shifted





over the low PI seasons and then increased again by 2012 (Figure 5). There was no difference seen between the average bottom temperatures over the whole spawning season as opposed to only the start of the spawning season. The winter months (June to August) had less than one degree of between year annual variability over the entire temporal scale.

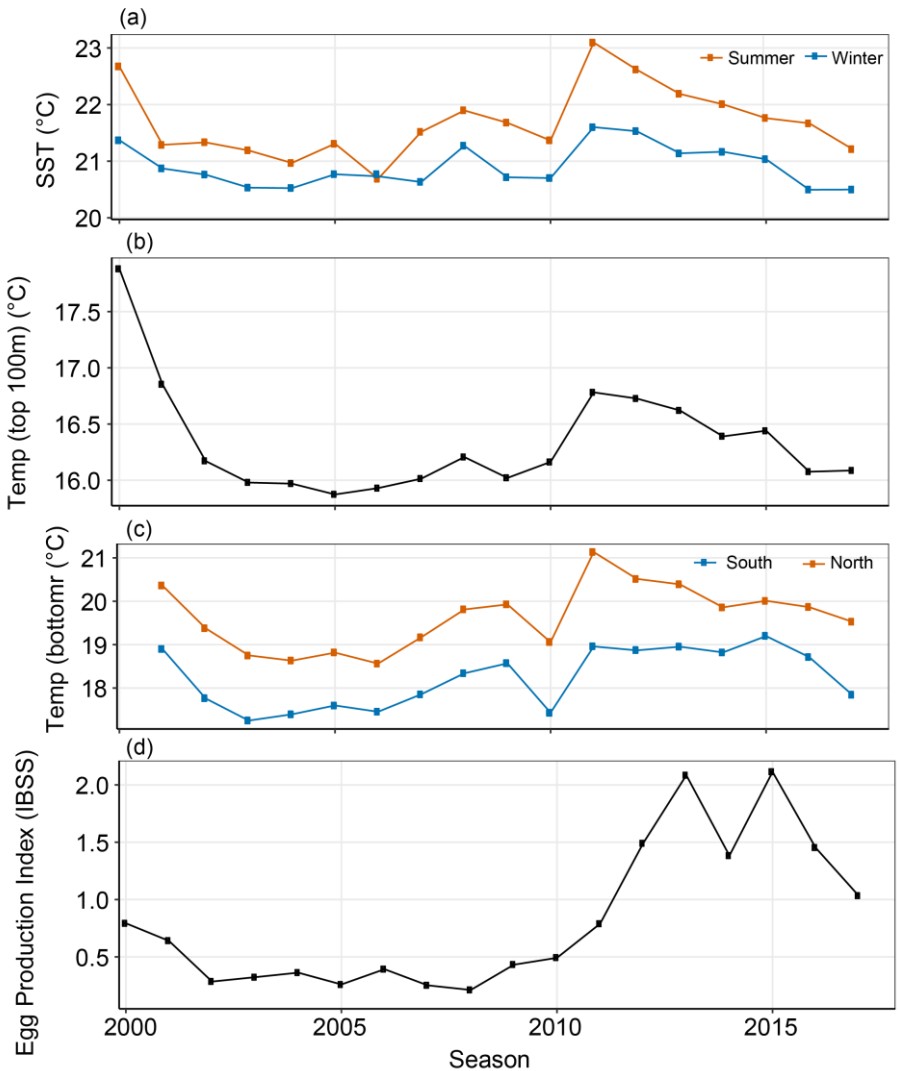

**Figure 5 Parameters calculated for seasonal analysis of puerulus index (PI) (a) Sea surface temperature (°C) from 24 - 34° S and out to 108° E obtained from the SSTAARS daily dataset on the IMOS AODN portal from 1996 to 2016; (b) Temperature in the top 100 m of the offshore area from 24 - 34° S and east of 108°E over Jan - Dec, the average time phyllosoma are offshore (c) Average bottom temperature (40 – 80 m depths) (°C) of the spawning season (season – 1) for associated PI season for October to March in the northern (blue line, shown in Figure 1a) and southern latitudes (red line, shown in Figure 1a) of the fishery (d) The Independent**
**Breeding Stock Survey Index (IBSS) lagged 1 year to give a spawning stock index for the year prior (de Lestang et al. 2016).**

The IBSS was consistently under 0.5 from 2002 until the 2011 season before increasing three-fold by 2013 to record-highs (Figure 4) (de Lestang et al., 2016). Previous studies have not found the IBSS to be implicated in the recruitment failure(de Lestang et al., 2015), but it has been included in the current analysis for completeness and because studies of





recruitment failures in other fisheries have frequently suggested spawning biomass to be a factor. Where the IBSS increased
from 2011, this is suspected to be due to the restrictive fisheries management, designed to preserve spawning biomass,
introduced after the recruitment failure.

The LC was the strongest over the winter months, reaching 7 Sv in 2000 at 34º S (Figure 6a). The LC in summer was
strongest in 2010, at 27º S which aligns with La Niña conditions but does not correspond to a maxima in winter strength (Figure
6b) (Wijeratne et al., 2018). The CC strength was highly variable between latitudes. Over the initial months of the current
forming (Figure 7a) it is, on average, strongest at 30º S. The CC displayed a roughly a similar pattern across all latitudes with
less variability in current at 27º S where it is weakest (Figure 7). CC minima occur over the 2010 to 2012 seasons in both the
early and late strength signals. Spatial variations in the LC and CC were clearly distinguishable with increased LC at the
southern latitude (Figure 6) and the strongest CC signature at 30º S (Figure 7) as reported previously by Wijeratne et al. (2018).

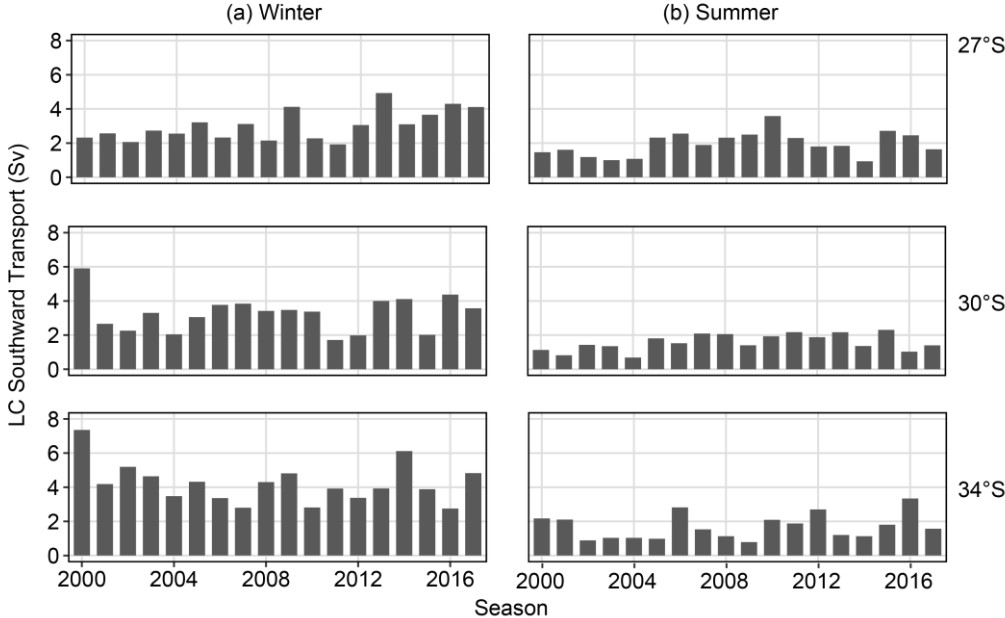

**Figure 6 The Leeuwin Current strength (southward transport, Sv) at 27º S, 30º S and 34º S over (a) winter (May - July) and (b) summer (December - February)**

Water circulation over all latitudes of the fishery were predominantly driven by the LC with the changing topography
down the coast causing less onshore flow on average within the centre of the fishery (29º S) (Rennie et al., 2007; Feng et al.,
2010; Wijeratne et al., 2018). Regions with a wider continental shelf generally have higher retention of waters therefore causing
less cross-shelf transport of water. Depth-averaged cross-shelf transport of water was predominantly onshore at both 33 and
29°S (Figure 8). This was not unexpected given the steep topography of the continental shelf and LC interactions. Average
monthly variations in cross-shelf transport indicated that between April and September onshore transport increased at 33°S
and 29°S, however decreased at 29°S and 27°S (Figure 8). Coastal geographic features increase the spatial heterogeneity over



the latitudes and additionally, how the LC interacts with the nearshore (Feng et al., 2010). In particular at 27°S on average
offshore transport was possibly due to more mixing and a wider continental shelf and increased mixing around Shark Bay and
with the contribution of the Ningaloo Current likely playing a role (Woo and Pattiaratchi, 2008).

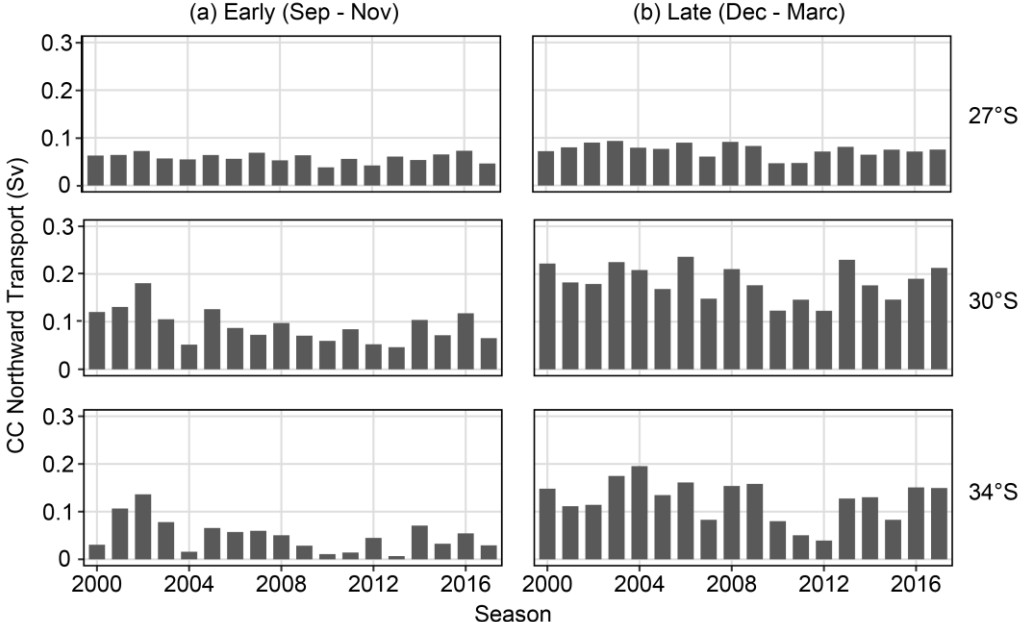

**Figure 7 The Capes Current strength (northward transport, Sv) at 27ºS, 30ºS and 34ºS over (a) the early portion of the summer
(September - October) and (b) late portion of the summer (December - March)**



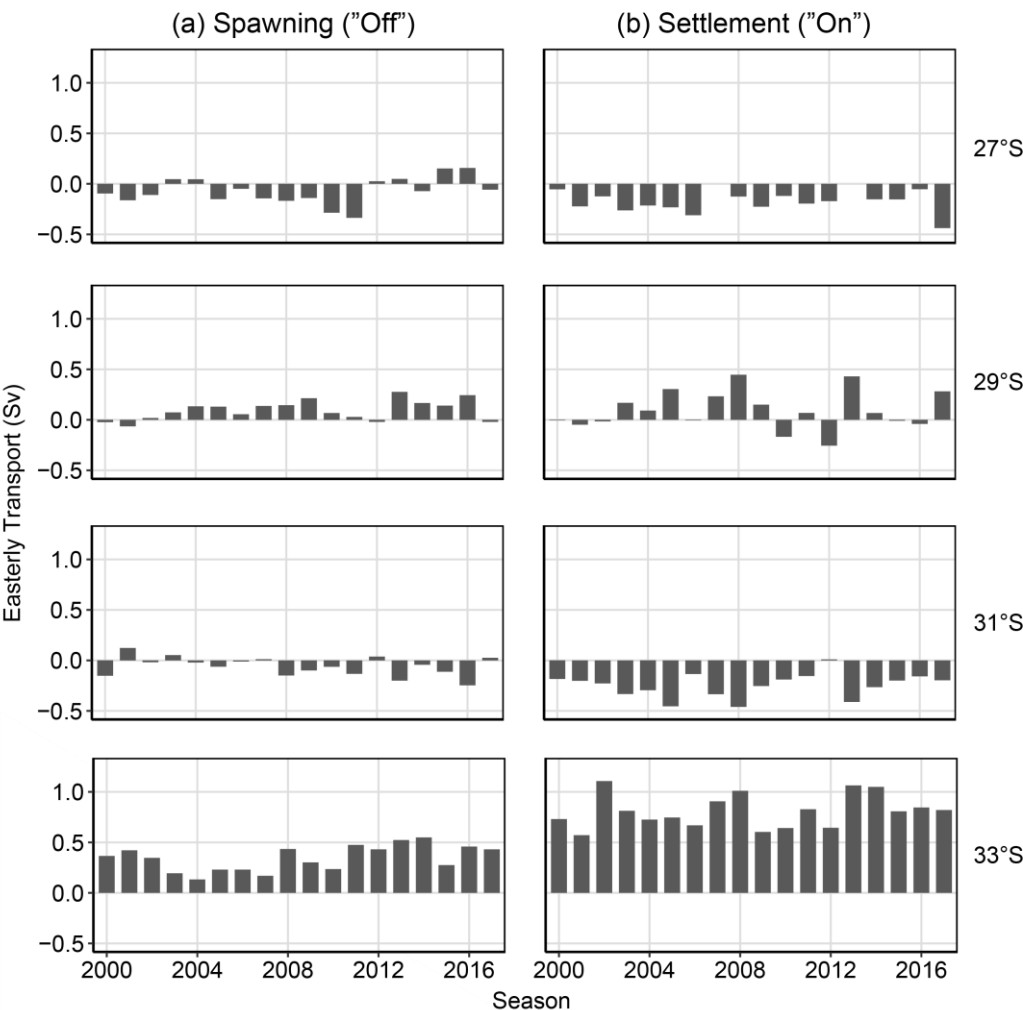

**Figure 8 Cross-shelf transport (easterly) between 200 - 50 meters over 2 degree latitudinal bins (26-28° S, 28-30° S, 30-32° S and 32-34° S). Averaged for the (a) spawning "off" transport season (September-February, season -1) and (b) settlement "on" transport season when variation in cross-shelf transport is the highest (April-September).**

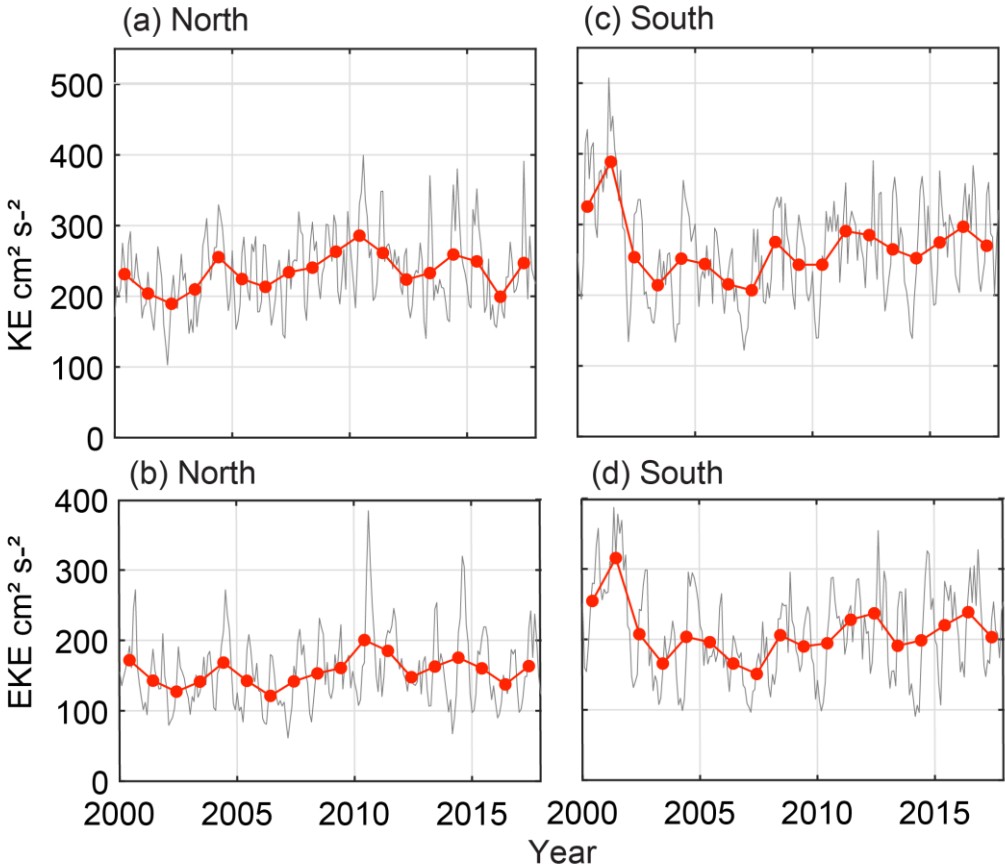

**Figure 9 Monthly kinetic energy and eddy kinetic energy (grey) with yearly averages (January-December) (red). North and south divide shown in Figure 1a. (a) KE (cm² s⁻²) in the north (b) EKE (cm² s⁻²) in the north (c) KE (cm² s⁻²) in the south (d) EKE (cm² s²) in the south.**

Variations in EKE and KE over the time-series show approximately 5-yearly patterns (Figure 9) in fluctuation with maxima in 2000, 2005 and 2011 aligning with ENSO events (Pattiaratchi & Siji 2020). The southern box (Figures 8c and d)

had an increased variability suggesting there was a greater variability in the LC at southern latitudes. Whereas, over the northern box, there was a peak in 2010 which was more pronounced in the KE than EKE (Figure 9). Over 2002 to 2008, the KE and EKE were both relatively low, indicating a weaker LC over those seasons.

## 4.2 Multiple regression analysis

Multiple regression analysis indicated that the factors which displayed the greatest relationships with early settlement

(Figure 10a), and were in line with our predictions (Table 1), were KE in the south (for two southern PI sites with some further importance at Lancelin and Alkimos), and both LC at 27° S during summer and early CC at 30° S for westward transport of phyllosoma (Jurien Bay, Lancelin (LC) and Port Gregory (CC)). The LC in winter and CC over the early summer, both at 27°


S, during puerulus returning, had a positive relationship to coastal sites north of Jurien Bay which was also predicted. Whereas for late settlement (Figure 10b), the dominant factors were the LC at 30º S during summer impeding the offshore transport of

larvae, the IBSS, the LC at 27 º S while puerulus were returning (Alkimos and Warnbro), and KE in the south at Lancelin and Alkimos.

**Figure 10 Variable importance scores within 2 AIC of the top model from the multiple linear regression analysis (GAM) to predict the (a) early and (b) late settlement at sites. The timeline (above) indicates the timing of the variables from either the spawning**
**season (s-1) where larvae are moving offshore (westward) and the settlement season (s) where larvae are moving eastward. Positive (red), zero (white) and negative (blue) relationships with variables are shown and variables within the most parsimonious model for each site are indicated (X).**

In contrast to our predictions (Table 1), sites were correlated by a negative relationship to KE in the north for early

settlement. This was stronger for the northern sites. A strong KE implies a strong LC signature over the defined spatial area

(Figure 1). Comparatively, EKE and KE in the south had positive relationships to sites in the centre of the fishery for early and late settlement suggesting a different forcing could be at play (Figure 10). The possibility that two adjacent parts of the south-east Indian Ocean would have opposing effects on settlement at only two locations appears spurious. However, the South





Indian Counter Current flows eastward within the defined southern 'box' of KE, one would expect if this had such an influence that it would be true for all sites and not only within the early portion of the season (Wijeratne et al., 2018). This difference in

KE relationships suggests different driving mechanisms over the fishery on both the temporal (early and late) and spatial (north and south) scale.

Interestingly, the LC during the spawning season at 27º S was within the most parsimonious model for Lancelin early settlement as a negative relationship (Figure 10a). This may be due to the LC being too strong for some early-stage phyllosoma to cross the shelf to the deeper ocean without being swept too far south for survival, this may explain why KE in the north is

showing a negative relationship. LC at 27º S over the spawning season, KE in the north and EKE in the south are the only few parameters where similar patterns occur in the early and late settlement (Figure 10 a & b). In particular, for the Abrolhos, KE in the north is within the most parsimonious model for early and late settlement. Due to the sites' location offshore, within the defined area of KE, it is possible that increased water movement prevents puerulus from successfully settling on the islands and instead they are transported or swim elsewhere with less resistance.

The model results provide some clues regarding the influence of the LC and CC (LC winter, LC summer, CC early and late) on phyllosoma when they are in later-life stages. A stronger LC in winter (puerulus reaching the continental shelf) and a stronger CC (puerulus settling on reefs) were good for early settlement at Dongara. This was a hypothesised result (Table 1) and was also consistent at adjoining sites (Port Gregory for LC and Jurien Bay for CC, Figure 10a) (Pearce and Pattiaratchi, 1999). However, for later settlement, the opposite relationships occur. The LC in summer has a negative relationship to Port

Gregory (and Abrolhos) and the CC has a negative relationship to Warnbro (Figure 10b). These results are in line with our predictions, with a northern site (Port Gregory) being negatively impacted by a strong LC causing southward advection and a southern site (Warnbro) being negatively impacted by a strong CC transporting puerulus northward, but these trends becomes vague over the central latitudes of the fishery where little to no relationships are found, especially during the summer of hatching (Figure 10a, CC 27º S early).  This draws attention to the spatial and oceanographic heterogeneity of the study sites.

Given the mix of both expected and unexpected and strong and weak results from the multiple regression analysis, particularly for the early settlement, it is clear complex forcing's are at play in the system, with both currents flowing in opposing directions perpendicular to the direction puerulus are swimming. The influence of factors found to have the strongest correlation on puerulus settlement is presented in the following section (4.3).

The IBSS shows a strong positive relationship with Port Gregory, Dongara, Jurien Bay, Lancelin and Warnbro over

the late settlement, however has little relationship with the early settlement (Figure 10 a & b). These positive relationships however, do corroborate that the egg production has an influence on the amount of larvae returning to the coast as puerulus and may more accurately represent the spawning stock of puerulus reaching the coast over the latter half of the settlement season. Given the longer time series available for the IBSS and Dongara settlement, we reanalysed the relationship and found that pre-2000 that IBSS is a reasonable predictor with an $R^2$ of 0.434 and explaining 30.3% of the deviance in the Dongara

settlement (Figure 11).  In an ideal scenario, one would expect all sites, both early and late to have a position relationship to




the spawning stock. For locations where increased IBSS did not have a positive correlation with PI it may be solely that the influence of oceanographic factors on the puerulus settlement was higher.

Cross-shelf transport shows sporadic patterns between the sites. Cross-shelf transport for the spawning season (27º S, 29º S and 33º S, Figure 9a) has some positive links to increased settlement in the early half of the season for sites south of

Dongara. However, this is the opposite for 31º S (Figure 10a) but these relationships are less pronounced over settlement later in the year (Figure 10b). Various transportation pathways are likely working in opposing directions to influence settlement at different locations. Average cross-shelf transport has previously indicated a break over the mid latitudes of WA (Wijeratne et al 2018, Figure 10).

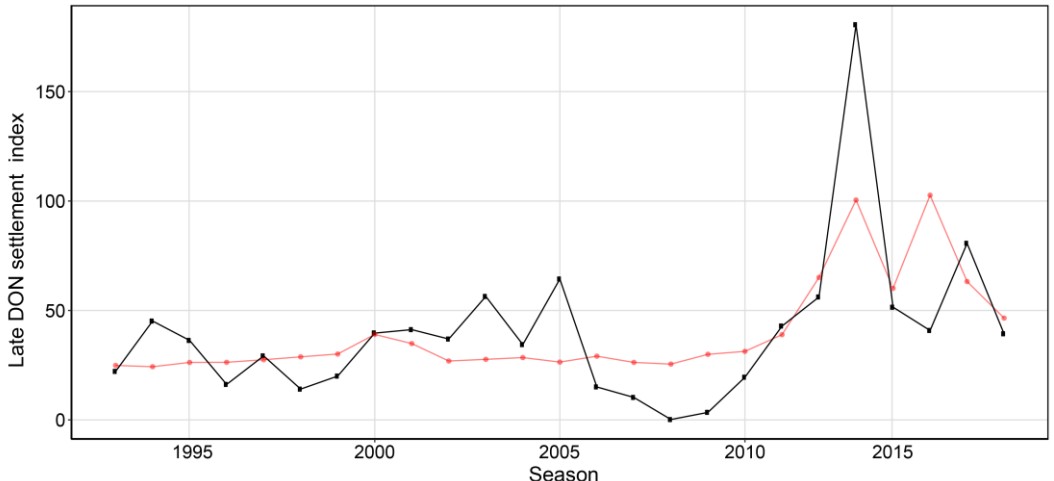

**Figure 11 Observed (black) and modelled (red) late Dongara (DON) settlement index from 1993 to 2018. The red line is the model parsimonious model extended to 1993.**

On an annual timescale, increased strength in KE in the northern offshore areas of the fishery (Figure 1) were associated with a decrease in puerulus settlement in the early portion of the season. It is not uncommon for the advective behaviour of large-scale eddies to negatively affect crustacean species (Medel et al., 2018; Nieto et al., 2014). Retention and

dispersal of larvae can also differ in persistent eddy scenarios where a more uniform shape likely leads to retention but a more eccentric shape leads to dispersal (Cetina-Heredia et al., 2019b). From our analysis we have not defined directionality and size of the LC eddies, but is an important consideration that would require further modelling, outside the scope of the current study.. Earlier studies however, suggest an increase in eddies positively influence the retention of larvae and therefore transport across the shelf and to the nearshore (Griffin et al., 2001; Yeung et al., 2001; Cetina-Heredia et al., 2019a, 2015). Here, we have used

the mean annual KE over a large spatial area and the site in question is north of the highest density of puerulus settlement. By using a large spatial area we have omitted the influence of sub-mesoscale features within the region which could impact a puerulus' ability to cross the shelf and reach coastal habitats (Cosoli et al., 2020).

These analyses suggest not only that PI at each site is influenced by different environmental drivers, but this also varies depending on when puerulus are returning to shore (early or late, Figures 10 9a and b). However, we can establish





discernible patterns with some certainty and physical relevance. Sites closer in latitude have similar results, Abrolhos and Cape Mentelle being the exception. Abrolhos is located off the shelf and has historically had different trends in PI compared to other locations and even on the adjacent coast. Also, since the recruitment failure of 2008 and 201109, PI has recovered to pre-failure levels whereas coastal locations have not, in particular at Lancelin and locations further south, the late season settlement has not recovered (Kolbusz et al., 2021). Cape Mentelle has historically had low settlement and also has a unique location

being the farthest south. Early settlement at Cape Mentelle also had little difference between all top models within 2 AIC of the most parsimonious model (Cross-shelf transport at 33⁰ S during spawning season). Despite one variable having the highest importance and being in the top model there was little difference between all models within 2 AICc units. Jurien Bay and Lancelin have similar results within the early settlement, but not later settlement.

Given the several months over which lobster larvae hatch, followed by their prolonged pelagic life cycle and

settlement estimated to occur some 9-11 months later (Phillips, 1981), the large amount of variation and lack of strong relationships between environmental or biological predictors and PI was not unexpected (Figure 2). However, we have revealed patterns up and down the coast suggesting that both biological and environmental predictors can have a strong and sometimes consistent influence on puerulus settlement for adjacent sites.

### 4.3 Variation in oceanographic conditions

All the oceanographic factors examined here have been suggested to have a direct influence on *P. cygnus* larvae at some point in their first year of life. However, the forcing and interactions between these environmental variables over time were too complex to examine in the above multiple regression analysis but may have as much influence on successful puerulus settlement as opposed to instantaneous values used. Over the large area of the fishery, the various oceanographic mechanisms are acting differently, sometimes in competition (Figures 9a and b), to provide contrasting results for the different sites. For

this reason, using the results of the above multiple regression analysis as an exploratory tool, we have additionally drawn upon patterns in the south-east Indian Ocean and WA coastal zone over the last two decades as complementary lagging conditions which alongside the fishery changes that may have contributed to the "worst case scenario" and resulted in the recruitment failure in 2008 - 2009.

### 4.3.1 Inter-annual: Hiatus period and cross-shelf transport

Over the past 30 years links between ENSO events and the puerulus index (PI) have been well documented through using the SOI (Figure 3) (Pearce and Phillips, 1988; Caputi et al., 2001). Warmer temperatures are experienced during stronger LC conditions evident during the La Niña phase (positive SOI), providing better conditions for larval development. The SOI record since these relationships began highlights the possibility of sustained neutral ENSO conditions to be a reason for this breakdown (Pattiaratchi and Hetzel, 2020; Pattiaratchi and Siji, 2020). From 2000 until 2009, both a moderate La Niña phase

nor moderate El Niño phase occurred and consequently the energy in the system also decreased (Figure 3). After 2009, an unusually strong LC (La Niña, 2011) was preceded by an unusually weak year (El Niño, 2010) (Huang and Feng, 2015). Prior





to 2008 there were fluctuations in LC strength, or phases of a moderate strength, over several seasons whilst the puerulus settlement also fluctuated is a similar fashion (Figure 3, and Caputi et al., 2001, Figure 5). An extended period (> 5 years) of low or neutral ENSO conditions had not yet been experienced since puerulus collection began therefore the relationship

breakdown is not surprising. Recovery in puerulus numbers began after the strong La Niña in 2011 taking a few seasons to reach levels prior to the hiatus. If these strong La Niña conditions had not occurred what would have been the response? Whether this 'delayed' recovery was due to memory in the system adjusting or the recruitment numbers recovering after management changes to fishing the spawning stock, increasing the IBSS is unsure.

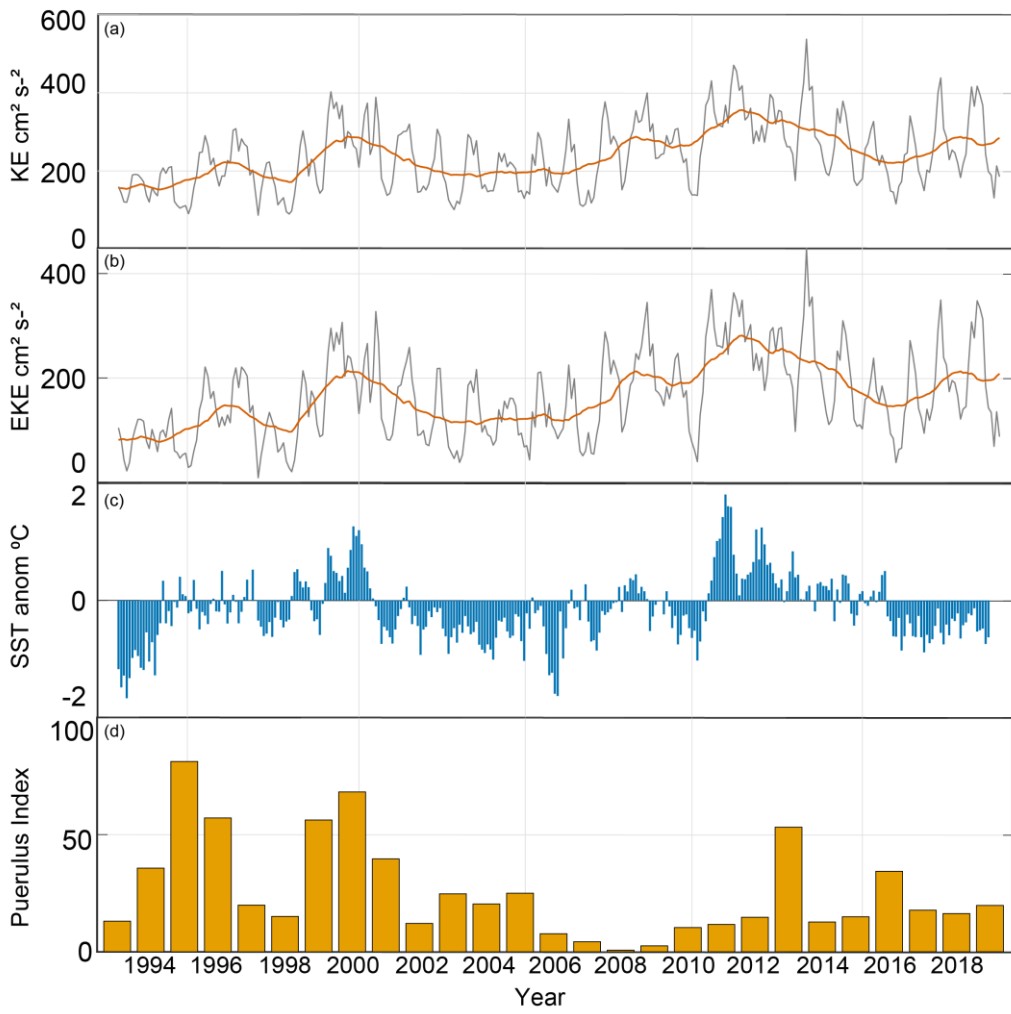

**Figure 12 Kinetic energy and eddy kinetic energy (cm² s⁻²) from altimeter data, the SST (ºC) anomaly from SSTAARS and Puerulus Index for the whole Western rock lobster fishery.**

Similarly, SST anomalies (Figure 12) have periods of neutral conditions and below average temperatures, respectively, from approximately 2000 to 2008 (Pattiaratchi and Hetzel, 2020). It is possible that these extended low activity





conditions have caused a shift in the conditions experienced by pelagic western rock lobster larvae. The patterns may be due

to atmospheric and oceanic processes that imprint themselves upon the SST field. The thermal energy of the ocean is

communicated to the atmosphere via the sea surface which is controlled by the SST and thus SST on a spatial scale plays a

key role in regulating climate and variability. The extended period of cooler SST anomalies may have contributed to the low

2008 and 2009 settlements. A decreasing PI over the start of the century was in line with these patterns and then recovery post

the maxima experiences with the 2011/12 La Niña (Boening et al., 2012) which perhaps took the system some years to return

to conditions as normal. It may not be a season specific factor which has caused the years of low settlement in the late 2000s

but rather consecutive years of these 'hiatus' conditions (Figure 12) which have driven a regime shift in the environment

impacting *P. cygnus* pelagic life stages (DeYoung et al., 2004).

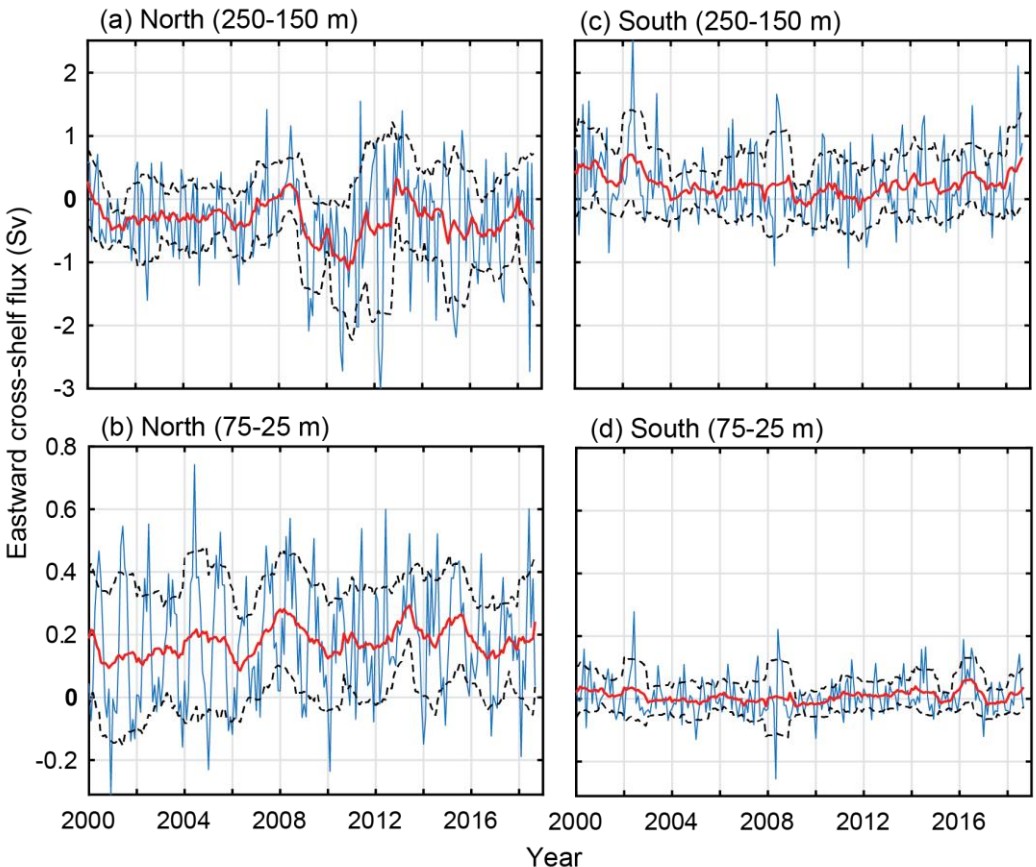

**Figure 13 Cross-shelf transport (easterly) between 2000 and 2018 off the shelf. North and south divide shown in Figure 1a. (a) North**
**cross-shelf flux between 250-150 m and (b) 75-25 m. (c) South cross-shelf flux between 250-150 m and (d) 75-25 m. The yearly moving**
**mean (red) and with the positive and negative moving standard deviation (dashed black) are included.**

The lack of statistical relationships with cross-shelf transport, despite it being undoubtedly a forcing mechanism

behind larval transport into the nearshore was unexpected. This is especially cause for further investigation since the 'hiatus'





conditions or shift after 2008 in the LC, one would expect some form of accompanying change in cross-shelf flux (Pattiaratchi
and Hetzel, 2020). Over the 50 m contour there is no noticeable inter-annual changes over the 18 years between the south or
north bins. In the south there is less variation in the standard deviation from the mean (Figure 13, dashed lines) however it is
predominantly onshore transport on the shelf (50 m) and over the continental shelf (200 m). The cross-shelf transport in the
north shows a clear increase in variability over the continental shelf (Figure 13, 200 m) after 2008, highlighting increased
movement in water over the northern part of the fishery (Figure 13). This is where the LC increases over the summer period
(Figure 5b) and a shift in puerulus settlement occurs.

### 4.3.1 Seasonal: Capes Current and Leeuwin Current interactions during summer

A shift in mean LC and CC conditions early in the season (late austral spring) has occurred subsequent to the
recruitment failure. Prior to 2008, conditions were fairly neutral over the early settlement season, leading to high/low puerulus
settlement to be potentially controlled by oceanographic and/or biological factors (Figure 14). Since 2009, conditions are such
that LC dominating (blue years skewed to the right, Figure 14) could give rise to average to low puerulus conditions across all
sites except at Cape Mentelle where higher settlement was experienced during these seasons, where Cape Mentelle is the most
southern of the sites and potentially most isolated from the LC. Comparatively over the late period of settlement, the system
is dominated by the CC with higher settlement occurring later in the summer (Figure 15). The years of recruitment failure both
have neutral conditions where neither the CC nor LC is particularly dominant over the later portion of the season. The same
exists for 2007 and 2008 during the early portion of the season. Interestingly, during these recruitment failure years, the spatial
extent of the LC during spring months (early settlement) was also larger or on par with the 6 months prior during  time when
newly hatched phyllosoma transport across the shelf into the open ocean may have also been impacted  (Huang and Feng,
2015).





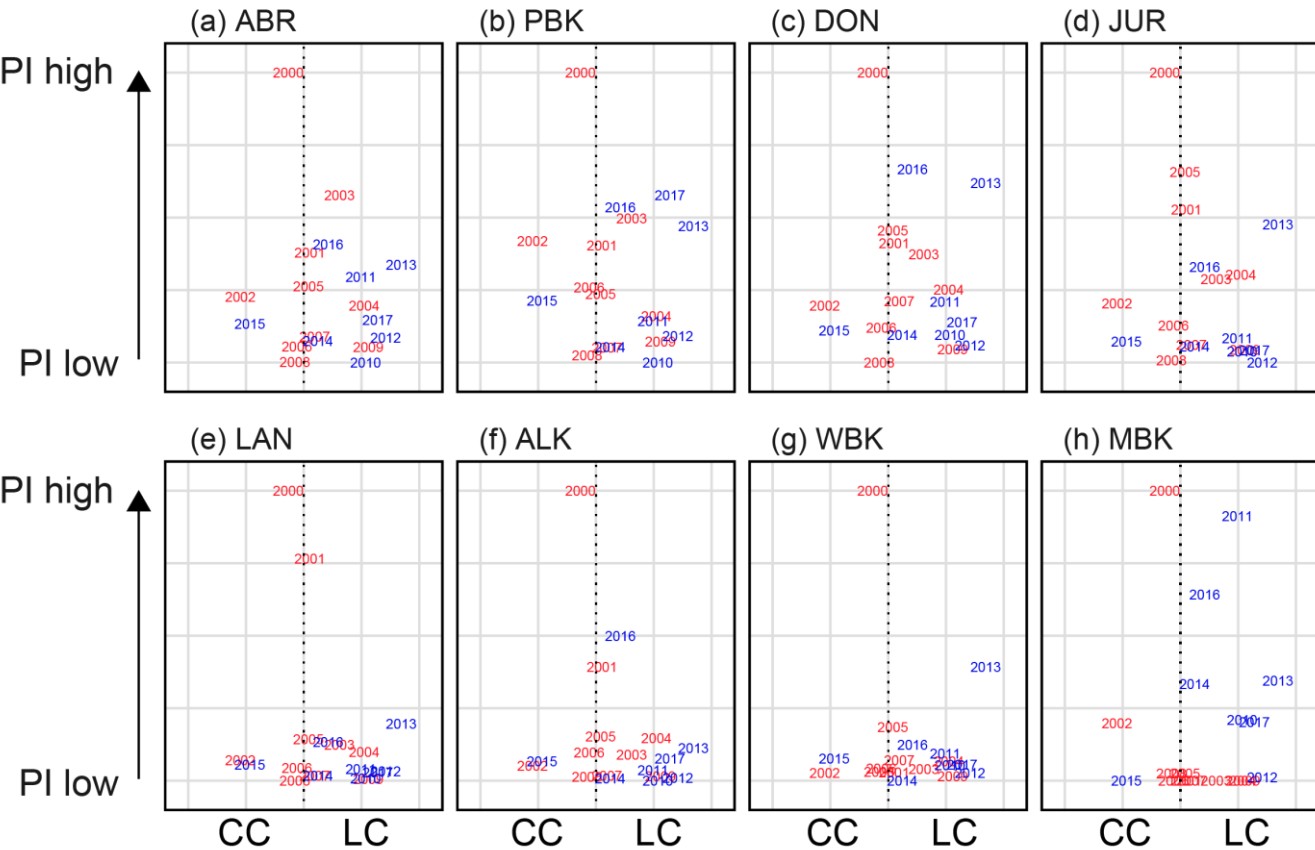

**Figure 14 Relationship of the more dominant current during the late spring/early summer (September - November) to early PI at monitoring sites (a) Abrolhos (b) Port Gregory (c) Dongara (d) Jurien Bay (e) Lancelin (f) Alkimos (g) Warnbro and (h) Cape Mentelle. The x-axis is the difference between the LC and CC standardised, providing an indication of which is more prominent at the time. Red indicates the seasons before the recruitment failure, blue indicates seasons after.**






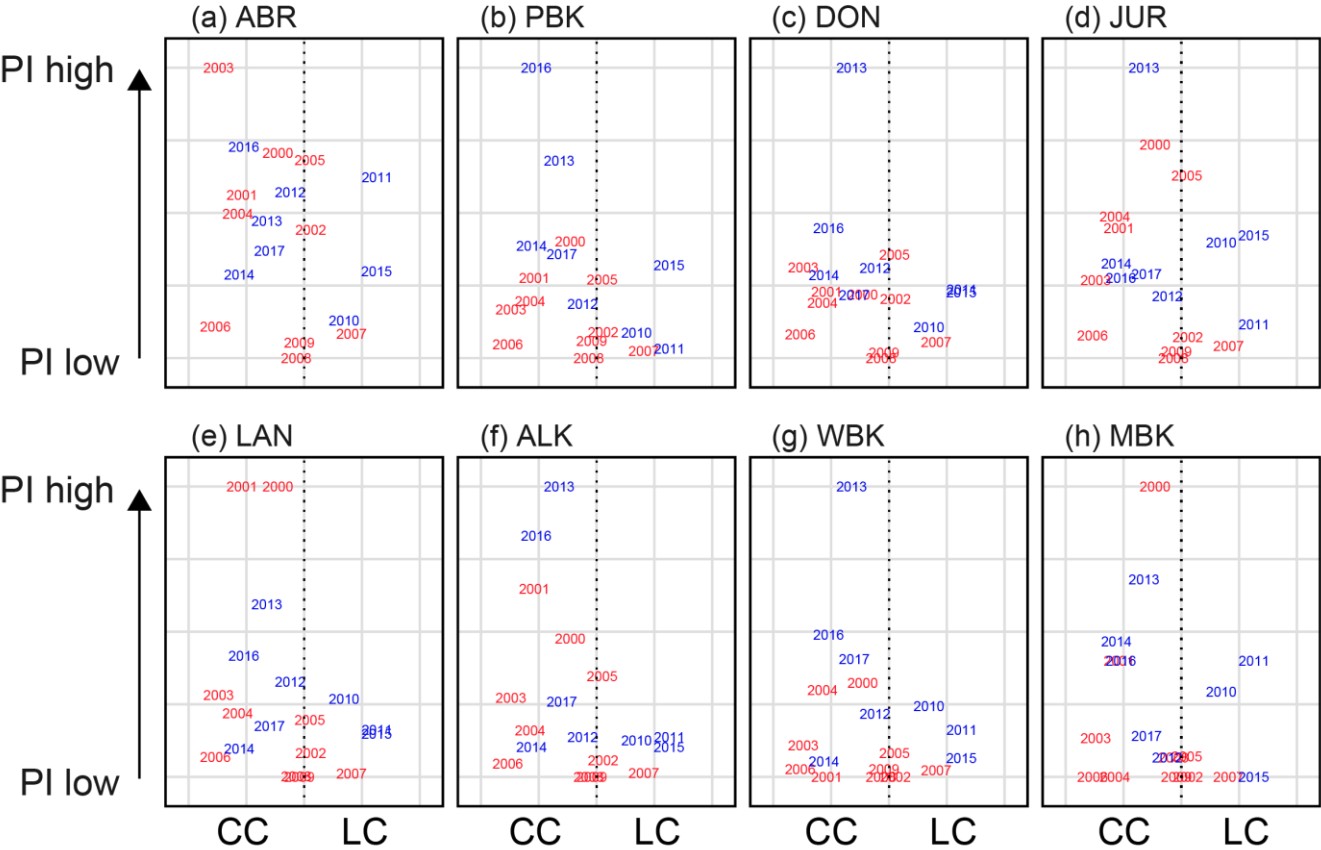

**Figure 15 Relationship of the more dominant current during the late summer/early autumn (December - March) to late PI at monitoring sites (a) Abrolhos (b) Port Gregory (c) Dongara (d) Jurien Bay (e) Lancelin (f) Alkimos (g) Warnbro and (h) Cape Mentelle. The x-axis is the difference between the LC and CC standardised, providing an indication of which is more prominent at the time. Red indicates the seasons before the recruitment failure, blue indicates seasons after.**

These patterns suggest that perhaps a timing mismatch is in play, suggested previously by de Lestang et al. (2015).

An increased LC while puerulus are crossing the shelf may transport them southward and either settling closer to Cape Mentelle or wash them too far offshore to return. In contrast, a strong CC is likely to assist transport northward along the shelf, hence the increase in settlement over the later part of the season. Interaction between these two important currents over the shelf is clearly influencing the movement of puerulus onshore and needs to be investigated in greater detail. Particle tracking modelling over these months would be a possible way to investigate these interactions, however this beyond the scope of the present

study.

## 5 Conclusions and Implications

The objective of the current study was to determine oceanographic and biological factors that may explain the previously unexplained failure of puerulus settlement (2008-2009) and subsequent change in the proportions of puerulus settling in early



vs late parts of the season (Kolbusz et al., 2021). This was completed through an exploratory multiple regression analysis

encompassing direct oceanographic and biological factors likely to influence the pelagic early-life cycle of *P. cygnus*. The main conclusions were as follows:

- Local oceanographic and biological conditions greatly influence the settlement of *P. cygnus*, as settlement of puerulus at adjacent sites along the coast tend to be influenced by similar oceanographic and biological variables. This is also true for the offshore Abrolhos Islands which had a unique yet consistent pattern of settlement that correlated with

particular oceanographic and biological factors.

- On a fishery-wide scale, the period of recruitment failure (2008 and 2009) and the associated low settlement period (2004 – 2010), coincided with a period of hiatus in the Leeuwin Current system. This was associated with mainly neutral ENSO conditions and slightly cooler SST anomalies.

- Increased KE in the northern region of the fishery was negatively correlated to puerulus settlement in the early period

of the season whilst the KE (and EKE) in the southern region was positively correlated at selected sites. This suggests different driving mechanisms over the whole range of latitudes that encompass the fishery for settlement early and later in the season.

- Seasonal variation of the LC system likely controls the conditions that favour increased puerulus settlement. A strong LC during the summer months of hatching, negatively affects puerulus settlement at central sites after their pelagic

phase. During the subsequent winter, the system is dominated by a strong southward LC and its associated eddies which then assists onshore phyllosoma transport. Then, whilst the newly metamorphosed puerulus are moving onshore across the LC, should it be too strong, the current can negatively impact settlement at the northern sites but positively impact the southern sites.

- An increase in the strength of the LC in the summer months since 2006 combined with a decrease in the strength of

the CC over the early summer months may have caused a timing mismatch for puerulus settling on nearshore reefs. If the LC was stronger in summer, a strong CC would be needed to counteract the southward flow to get the puerulus transported northward, which has occurred in recent years. The CC in the latter half of the summer however, has been less variable and not declined to the same extent, potentially explaining the trend of greater settlement later in the season.

- Between 2008 and 2011, the cross-shelf transport in the northern half of the fishery across the continental shelf became more variable whilst consistently offshore. This overlap with the period of recruitment failure suggests that cross-shelf transport in an offshore direction could have reduced the transport and subsequent settlement of puerulus onshore.

These findings have implications for fisheries management and their modelling of future stocks of catchable western rock lobsters. Where environmental conditions are suspected to alter the puerulus settlement, these can be incorporated in





planning and management. For example, reductions in fishing effort can encourage an increase in larvae for the following season (de Lestang et al., 2015).

Despite its exploratory nature, this study offers insight into factors potentially behind the recruitment failure which have not been addressed prior. It also expands our current understanding of what oceanographic variables potentially influence puerulus settlement and how the variables themselves are intertwined in this complex system. With the available numerical modelled data we are able to show that it is not solely the LC which is the dominating factor behind puerulus settlement variability but the CC, cross-shelf flow and the state of the system as a whole.





**Appendix A**

**Table A1. Oceanographic data sources and sites used to calculate the 40 predictor variables included in time series analysis. Unless stated, the timing is within the season of puerulus settlement. Spawning season indicates that the variable is from the year prior or within the spawning season as larvae hatch.**

| Predictor Variable | Data source | Northern | | Central | | Southern | No. Variables |
|---|---|---|---|---|---|---|---|
| **Leeuwin Current strength** (Austral winter mean) | ozROMS | 27ºS | | 30ºS | | 34º S | 3 |
| **Leeuwin Current strength** (Austral summer mean) Spawning season | ozROMS | 27ºS | | 30ºS | | 34º S | 3 |
| **Leeuwin Current strength** (Austral summer mean) | ozROMS | 27ºS | | 30ºS | | 34º S | 3 |
| **Capes Current strength** (Austral spring mean) | ozROMS | 27ºS | | 30ºS | | 34ºS | 3 |
| **Capes Current strength** (Austral summer mean) | ozROMS | 27ºS | | 30ºS | | 34ºS | 3 |
| **Capes Current strength** (Austral spring mean) Spawning season | ozROMS | 27ºS | | 30ºS | | 34ºS | 3 |
| **Capes Current strength** (Austral summer mean) Spawning season | ozROMS | 27ºS | | 30ºS | | 34ºS | 3 |
| **Cross-shelf transport** – offshore (spawning season) 150 – 50 m (September - March mean) | ozROMS | 26-28ºS | 28-30ºS | 30-32º S | 33-33ºS | | 4 |
| **Cross-shelf transport** – onshore 150 – 50 m (April - September mean) | ozROMS | 26-28ºS | 28-30ºS | 30-32º S | 33-33ºS | | 4 |
| **Bottom temperature** (October - March) Spawning season | ozROMS | 24.5-30.5°S | | | 30.5-34.5°S | | 2 |
| **EKE** (January – December mean) | ozROMS | 24.5-30.5°S | | | 30.5-34.5°S | | 2 |
| **KE** (January – December mean) | ozROMS | 24.5 - 30.5°S | | | 30.5-34.5°S | | 2 |
| **SST** October – March mean (summer) April - September mean (winter) | SSTARS, Integrated Marine Observing System (IMOS, oceancurrent.imos.org.au) Australian Ocean Data Network (AODN) portal (portal.aodn.org.au) (Wijffels et al. 2018) | WA waters extent 21ºS-36ºS 108ºE to coastline | | | | | 2 |
| **Temperature** in the top 100 m | ozROMS | WA waters extent 21ºS-36ºS | | | | | 1 |





| (January – December mean) | | 108ºE to coastline | |
|---|---|---|---|
| **Independent Breading Stock Survey** (IBSS) Spawning season | DPIRD | Index for whole fishery | 1 |
| | | Total predictor variables | 39 |




**Data Availability**

The dataset for this research and relevant contacts can be found through the Department of Primary Industries and Regional Development website. http://www.fish.wa.gov.au/Species/Rock-Lobster/Lobster-Management/Pages/Puerulus-Settlement-Index.aspx

The ozROMS numerical model outputs are available at this THREDDS server http://130.95.29.56:8080/thredds/catalog.html

**Author contribution**

JK completed the analysis of the data and conclusions under the guidance of CH, TL, and SdL. JK wrote the manuscript and produced the figures with the help and inputs from all co-authors. TL developed the code for the multiple regression analysis (Fisher et al. 2016). All authors contributed to the article and approved the submitted version.

**Competing interests**

The authors declare that they have no conflict of interest

**Acknowledgements**

We would like to acknowledge the Department for Primary Industries and Regional Development (Fisheries) for the use of raw puerulus collector data. Thank you to Rebecca Fisher for her assistance with the multiple regression analysis toolbox.
Thank you to Sarath Wijeratne for ozROMS data processing assistance.





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
