# Peer review of "Using an oceanographic model to investigate the mystery of the missing puerulus"

_Biogeosciences, 2021_

## Author Response (AR1)

We want to thank this reviewer who has provided a detailed review of our manuscript and provided valuable comments and suggestions, which have significantly helped improve our manuscript.
We have done our best to account for these comments and have responded individually (red text) to each of them below.

The significant changes of the revised manuscript will include:
- *A simplification of the introduction and removal of repetitive information. Including a focus on the aims of the paper.*
- *Rephrasing of the results/discussion section in line with the concluding remarks*
- *A more succinct explanation of the weakened conditions in the Southeast Indian Ocean likely contributing to the recruitment failure*
- *Clearer explanation of 'season' and puerulus settlement with consistent phrasing*
- *Rephrasing of the results/discussion section in line with the concluding remarks*

We hope that our response and modification to the manuscript will satisfy the reviewers and editor so that our manuscript will be recommended for publication.

Kind Regards

Jessica Kolbusz and co authors

Anonymous Referee #1
**General comments**

This paper investigates several oceanographic factors and one biological factor to try and explain the reasons for the breakdown of the positively correlated relationship between western rock lobster puerulus settlement and the strength of the Leeuwin Current off WA. The authors employ a multiple regression analysis to predict puerulus settlement, and also explore the influence of seasonal and inter-annual variations in the Leeuwin Current and Capes Current.
Given the high economic value placed on the western rock lobster fishery in WA, and given the past predictability of future catch by 3-4 years in advance has played a significant role in the sustainability of the fishery, this paper explores new drivers that would benefit scientists and managers in the fisheries and biological oceanography space.

Thank you for the supportive comment!

To arrive at these findings, the paper presents a lot of information, particularly around the oceanography, which I think could do with further refinement in order for the main takeaway messages to be clearer. These findings are presented more clearly in the concluding points of the paper, but not so much in the results/discussion section.
Agreed. We will revise the manuscript accordingly to better illustrate the conclusions within the results/discussion section.

There is a lot of information presented in this paper which makes it quite tricky to digest in some parts. In particular, the high number of predictor variables and oceanographic patterns explored for each station, on top of splitting the data temporally and spatially. It's potentially borderline too much for one paper, but I understand the complexity of oceanographic and biological patterns and the need to examine the extent of variables chosen in this paper.
Yes, thank you for understanding the complex problem we, too, were faced with. Hopefully, the conclusions will be more straightforward by revising the areas you and the other reviewer have brought to our attention.

In general, some sentences are quite long and could benefit from a grammar check and the addition of commas. I strongly recommend another round of editing on the document. Clear, concise and grammatically correct sentences will help with the digestion of a vast amount of information.

Yes, agreed. A thorough grammar check has now occurred.

**Specific comments**
*Abstract*
Summarises the paper well

Thank you for the supportive comment!

*Introduction*
Overall, I think the information presented in the introduction is relevant but could be more streamlined, have a clearer flow and avoid repetition. The aim is also mentioned twice but with different wording, so I suggest to just provide it once in the last paragraph of the intro
Agreed – this has been modified in the revised manuscript.

Line 36: 'Puerulus settlement has subsequently recovered, but despite extensive research', can you give a quick indication of where the extensive research focused e.g., in the oceanographic space, biology space
As suggested below, with Line 96, this has been moved up to this location.

Figure 1: consider whether should be in the methods section, given the caption is also methods related.
An interesting comment. It has been moved to the 3.1 Puerulus settlement data section, but it is referred to throughout the Study Region section. It is subsequently now Figure 2.

Line 96: Could this sentence 'Prior to current study, research on the 2008/09 decline had included an examination of overfishing of the spawning stock (de Lestang et al., 2015) and whether conditions of survival were no longer met in the early pelagic life stages (Säwström et al., 2014)' be moved up to plug the missing info mentioned in comment above Line 36?
Yes – this has been moved to line 36

*Study region*
Line 150: Sentence starting with 'An increasing' needs rewording, sounds confusing. Is 'replicated' the right word?
This has been changed as the information is also repetitive in line 129. This final sentence is now -
As the LC strength increases, as does its associated eddies and meanders.

*Methods*
Line 158: 'The fishery-wide standardized puerulus index, PI, is calculated based on the seasonal (May – April) mean puerulus settlement numbers from all 8 sites, then summed to obtain an annual index (Kolbusz et al., 2021)' repetition. Already mentioned in Intro. Pick one or the other. Suggest keep in methods.
Removed from the Introduction.

Line 236: Can you clarify this sentence 'Models containing variable combinations with correlations > 0.4 were excluded, to eliminate potential problems with collinearity and overfitting (Graham 2003)'? Are you referring to predictor variable combinations? If so, isn't the cut off 0.4 correlation between two predictor variable quite low? Typically it is set at 0.7

or 0.8 before you would start to exclude correlating predictor variables. But I can see further down you mention >0.8. So perhaps just needs clarifying a bit more.

*Yes – we must not have changed the value in the text – this was kept at 0.7. The 0.8 further down is regarding before completing the GAM analysis. This has been moved to the second paragraph of this section and rephrased to be more explicit.*

Table 1: 'Predictor **variables** and metrics'
*Changed*

Line 267: expand on what you mean by 'hiatus' and during what years?
*By 'hiatus' we mean an extended period of low activity in the SE Indian Ocean, namely the LC. This is evident between 2001 and 2007 (Pattiaratchi & Sij 2020). We agree this needs to be expanded on within the methods and aims. The exploration of long-term oceanographic conditions are explored due to processes in the ocean not having instantaneous changes. This is also relevant to your point re Line 457 and 'memory' in the system. This needs to be better explained by us. It will be in the following revision.*

*Pattiaratchi, C. and Siji, P.: Variability in ocean currents around Australia: State and trends of Australia's oceans report, Hobart, 1.4.1-1.4.6 pp., 2020.*

*Results/discussion*
Line 274: you say 'time-series patterns of the **spatial and temporal variability of the physical environment** experienced by P. cygnus larvae between 2000 and 2017', but then start of section 4.1 talking about variability in puerulus settlement. So summary wording around this section (from line 274) needs to be revised to reflect content accurately.
*Results and discussion are combined into three sections (1) time-series exploration of P. cygnus settlement between 2000 and 2017 and associated oceanographic conditions experienced by the larvae (when data was available); (2) exploring correlations of the oceanographic conditions with settlement data through multiple regression analysis, and (3) inter-annual and seasonal oceanographic variability.*

Line 275: '(2) exploring correlation of oceanographic conditions with multiple regression analysis'. Is that the right wording? Would 'correlations between predictor variables and puerulus settlement' be more accurate?
*Yes, this has been changed.*

Line 309: these sentences sounds contradictory: 'The CC strength was highly variable between latitudes. Over the initial months of the current forming (Figure 7a) it is, on average, strongest at 30â● ° S. The CC displayed a roughly a similar pattern across all latitudes with less variability in current at 27â□° S where it is weakest (Figure 7).'
*Agreed, this has been changed.*

Line 324: is it meant to say 'decreased at **31S** and 27S?' Not 29S?
*Yes, now changed, thank you.*

Line 412: suggest putting sentence 'From our analysis we have not defined directionality and size of the LC eddies, but **it** is an important consideration that would require further modelling, outside the scope of the current study' in method section as I was expecting you to mention anticyclonic or cyclonic eddies somewhere. It is an important consideration, and one I would have expected to see in the modelling/analysis given the detail of the other predictor variables used, and the potential impacts of the eddy types on body condition of larvae.
*Thank you for this comment. We have moved this sentence to where KE and EKE are defined.*

*For our analysis we have not defined directionality and size of eddies, but it is an important consideration pertinent to larvae energy stores that would require further modelling outside the scope of the current study.*

The paragraph starting at line 407 has been rephrased with this sentence removed.

Line 457: what do you mean by 'due to memory in the system adjusting'?
By 'memory in the system adjusting', we mean the climate inertia of the system. So the system would have some resistance or lag to changes such as KE and the extended cool water period. We have rephrased this to explain our findings better.

Line 466: is 'communicated' the right choice of word?
This has been changed to 'transferred.'

Line 487: 'A shift in mean LC and CC conditions' – clarify what conditions, speed, direction?
This sentence has been deleted since it is waffle and explained within the paragraph.

**Technical corrections**
Line 33: reword sentence to say 'During the 2008 and 2009 settlement seasons (May - April) there was an unexpected **settlement failure**, given the strong Leeuwin Current over those years.' In some cases throughout the document, you break up the sentence in a way that it doesn't flow. Such as the above example.

Agreed – changed. This has been changed to
*During the 2008 and 2009 settlement seasons (May - April), there was an unexpected recruitment failure.*

Line 52: missing bracket. Plus sentence needs clarification e.g.: 'The onshore transport and movement of **puerulus** across the continental shelf occurs mainly during August – January (late austral winter-summer), **where settlement occurs** in shallow areas of generally less than 5 metres depth'
Changed
*The onshore transport and movement of puerulus across the continental shelf peaks between September and February each year. Therefore, the circulation patterns of the south-east Indian Ocean influence spatially varied cross-shelf transport of the puerulus (Caputi, 2008; Feng et al. 2011). Puerulus settlement usually occurs in shallow areas of reef and seagrass habitats.*

Line 61: 'since **the** majority'
Added

Line 96: put month range for 'second half of the season' as did for first half of season
Changed

Line 105: 'effects are at play For example' missing full stop
Added

Line 135: join string of references 'Wijeratne et al., 2018) (Smith et al. 1991…'
Joined

Line 148: 'Eddies in LC are' – 'from the' or 'in the'
In LC is removed

Line 159: switch between 8 sites and eight sites – consistency. Spell out if between 1-10.

Changed to 'eight' throughout

Line 220. Missing full stop at end of sentence
Added

Line 270: why is '(Fremantle Mean Sea Level; Southern Oscillation Index)' needed in brackets when already defined in same sentence?
This was an issue with the referencing – it has been changed.

Line 320: add °S after 33 as well
Added

Line 324: Another example of a sentence that needs better grammar: 'In particular at 27°S on average offshore transport was possibly due to more mixing and a wider continental shelf and increased mixing around Shark Bay and with the contribution of the Ningaloo Current likely playing a role (Woo and Pattiaratchi, 2008).'
Rephrased.
In particular, at 27°S average offshore transport was possibly due to more mixing and a wider continental shelf. This is due to the topography of Shark Bay and the contribution of the Ningaloo Current likely playing a role (Woo and Pattiaratchi, 2008).

Line 362: Another example of a sentence that needs better grammar: 'However, the South Indian Counter Current flows eastward within the defined southern 'box' of KE, one would expect if this had such an influence that it would be true for all sites and not only within the early portion of the season (Wijeratne et al., 2018).'
We have expanded this with further clarity.
*However, the southern and central flows of the South Indian Counter Current (sSICC, cSICC) flow eastward respectfully within the South and North KE 'boxes' (Menezes et al. 2014). These current jets connect with the LC and may cause the temporal and spatial differences in KE and subsequent influences on puerulus settlement.*

Menezes, V. V, Phillips, H. E., Schiller, A., Bindoff, N. L., Domingues, C. M., & Vianna, M. L. (2014). South Indian Countercurrent and associated fronts. *Journal of Geophsical Research: Oceans, 119,* 6763–6791. https://doi.org/doi:10.1002/2014JC010076

Line 369: 'for survival**. This** may explain'
Changed

Line 382: 'but these trends becomes' – become
Changed

Line 395: positive not position
Rephrased

Line 405: most not model
Changed

Line 419: (early or late, Figures **10, 9a** and b)
Fixed
Line: 422: fix 'failure of 2008 and 201109'
Fixed

Line 453: 'fluctuated **in** a similar'
Changed

Line 510: Grammar: 'An increased LC while puerulus are crossing the shelf may transport them southward and either settling closer to Cape Mentelle or wash them too far offshore to return'
Rephrased
While puerulus are crossing the shelf, an increased LC may transport them past Cape Mentelle, away from suitable habitats.

Line 514: however this **is** beyond the scope

Changed

Anonymous Referee #1
**General comments**

This paper explores the influence of factors associated with the Leeuwin current (LC) and the Capes current (CC) in explaining the breakdown of the historical relationship between recruitment of western rock lobster recruitment and the strength of the LC. To do so, they use a series of GAMs to model the best predictors of the settlement index at 8 sites and pars of the settlement season. The predictors explored are associated with oceanographic parameters describing the LC and CC, as well as cross shelf transport and a breeding stock index.
Overall, this is a useful paper that will be useful to rock lobster fishery managers. However, because the paper is exploratory, there is a lot of complex information and the main goals and takeaway point a get lost.
Thank you for the comment. Significant re-wording includes a grammar update throughout the results/discussion section for takeaway points to be made clear. In addition, repetitive information in the introduction to make the paper's goals more explicit are also included.

The work itself appears to be solid, but is presented a little unclearly. I think a revision paying close attention to outlining specific goals in the introduction and structuring the paper around those goals could help.
Agreed. We will revise the manuscript accordingly to better illustrate the conclusions within the results/discussion section and importantly cut down and be more apparent within the introduction. This was also noted by Reviewer 1.

I also recommend revision for both grammar and to improve readability by making the writing more concise. Yes, agreed. A thorough grammar check has now occurred.

SPECIFIC COMMENTS
The introduction does a nice job of introducing the study system, but it does not introduce the impetus for the hypotheses well.
For instance, why did the authors choose to explore specifically KE and EKE (lines 116-117). This is discussed in the results/discussion, but is not introduced. I want to be convinced that these are useful parameters to explore—not simply that they are ones that can be examined. This could be a way to set up more specific aims of the paper.
Thank you for this critical comment. We completely agree. A driver of this research was to use direct measurements that are known to influence the *P. cygnus* early life cycle. KE and EKE are likely contributors to their early life cycle. We have now included this more thorough within the introduction.

Similarly, the authors have chosen to report a combined results and discussion section. I actually think the paper may be more clear if these are separate—there were many

paragraph in this section that were simply reporting of results with no discussion. This made the main points of the discussion get a little lost.

Thank you for this comment. Throughout intitial drafts, the Results and Discussion sections were separate. As the work was revised and complementary details added, we felt it necessary to combine the sections. Specific components of the work lead to the other, hence without discussing and interpreting the results of the GAMs, certain sections' results would be presented with no reason why. For example, the section regarding the LC and CC interactions would not fit in. We have gone through the results/discussion to better bring discussion points out in the text.

1. Figure 2: a set of photos would be better. Additionally, the figure does not capture what is written in the text well. For example, the 9 stages are not noted and the timelines indicated on the figure and in the text (line 51) do not match well. These also do not match the methods well (ex: line 182/183).

   A good point. The nine stages do not come into this work. Therefore we have modified the text referring to it. In addition, we have changed the images.
   We have changed the KE and EKE reasoning re their time offshore to be:
   *The months that phyllosoma are offshore, depending on when they have hatched, can be between October (year – 1) to March (year + 1) (Figure 2). The calendar year from January to December was used to obtain an average for the offshore conditions spanning the possible time frame offshore.*

2. Line 207- 208: I am unclear what you mean by the temperature of the top 100m of the assumed phyllosoma distribution. Does this mean the top 100 m of the water column (and that is the assumed distribution), or some other depth that is the top 100 m of their deeper distribution. Also, is there a reference for this assumed distribution or is this your assumption?

   Their distribution is known to be in the top 100 m of the water column. We will add this reference to the methods. Changed to:
   *The temperature in the top 100 m of the water column, east of 108ºE, as a mean annual value from ozROMS was also included. This accounts for temperature variation over the migrating depths phyllosoma occupy over their early pelagic life-cycle (Griffin et al. 2001; Feng et al. 2011).*

   Griffin, D., Wilkin, J., Chubb, C., Pearce, A., & Caputi, N. (2001). Ocean currents and the larval phase of Australian western rock lobster, *Panulirus cygnus*. *Marine and Freshwater Research*, *52*(8), 1187–1199. https://doi.org/10.1071/MF01181
   Feng, M., Caputi, N., Penn, J., Slawinski, D., de Lestang, S., Weller, E., & Pearce, A. (2011). Ocean circulation, stokes drift, and connectivity of western rock lobster (*Panulirus cygnus*) population. *Canadian Journal of Fisheries and Aquatic Sciences*, *68*(7), 1182–1196. https://doi.org/10.1139/f2011-065

3. Section 3.4 My understanding is that the authors fitted 16 different GAMs. I would call this section "generalized additive modeling" or something similar instead. It is also not clear what the response variables in the 16 models actually are though. The results (fig 10) say "settlement"—I assume this is the purlieus index? (it is clear that this is both for the late and early season and all sites in the figure and the methods).

   Yes, this change will be made to the title of that section.
   The response variables are termed puerulus settlement. The response variable is only the settlement from the early or late portion of the season, hence with 8 variables that are 16 GAMs in total. Therefore, we will remove the mention of the

puerulus index, commonly known as the value of puerulus settlement for the whole season and the whole fishery.
Continuity between puerulus counts/index/recruitment/settlement will be implemented and changed to only puerulus settlement.

Similarly, I am confused by why the GAMs were limited to linear relationships (line 238). Why use GAMs then? Can you please clarify?
Thank you for the comment. Cross-shelf transport was fitted as a three-knot spline (line 238). We used GAMs as we had the option to include them as linear or non-linear. The variation in results when all variables were included as non-linear was minimal, aside from cross-shelf transport. Therefore for ease of interpretation, our final results were completed using linear fitting.

4. Figure 5: this could be labeled more clearly. First, the x axis appears to be year, not season. Second, it would be more clear to have they axis of b labeled as "offshore temp (top 100 m). In the figure heading, I am unclear about how that season is defined as January – December. Isn't that the whole year? Finally, I suggest labeling the y axis for c as "Spawning Season Temp (bottom)."

Thank you for the comment. The season here referred to the puerulus season and associated PI, which is May – April, therefore also 12 months and is noted in the caption. However, we have not defined this effectively for the reader and, therefore, will better define the language surrounding this.
We will change the label for Figure 5c.

Fig 5-9: I also think it would be useful to shade the years of low PI as in other figures.
Yes, a fair point. We will do this.

5. While I quite like Figure 10, you need to have the details of the models and model selection process in an appendix at the very least.
Thank you for the supportive comment. Yes, we will add this to an appendix. We were initially unsure whether it was necessary.

TECHNICAL COMMENTS
- Figure 1a Legend—third sentence is incomplete
  This was also noted by reviewer 1 – we will edit this.
- The entire manuscript should be edited for comma rules to improve readability. I saw comma mistakes throughout that made sentences hard to read.
  This was noted by reviewer 1, and there has been a thorough overhaul of the grammar.
- Extraneous parentheses in line 135
  Deleted
- Table 1: I don't understand the coding in the hypothesized relationship column.
  The – and + signs were aimed to denote the positive or negative response of the puerulus settlement to the predictor variable in question. Therefore, this has been added to the Table caption.
- Fig 4; the abbreviations (ABR, PBK, etc) should be defined in the figure legend. Are these arranged by latitude? This could be included as well.
  Yes, these are arranged by latitude (north to south). We will include this in the figure caption
- Line 303-305 is an odd sentence structure

> Yes agreed. This has been rephrased to:
> Line 293
> *In addition, the IBSS increased from 2011; this is likely due to the restrictive fisheries management. After the lower than expected puerulus settlement in 2008 and 2009, restrictions to fishing were designed to preserve spawning biomass. Therefore the IBSS was expected to increase.*

- Figure 6-8: Here the x axes are year, not season.
  > Thank you for pointing this out. The values are referring to those used for the associated puerulus settlement season. Therefore season is the correct x-axes. For example, the bottom temperature value used for the 2001 settlement is the average temperature for October 2000 to March 2001 as this was when larvae were hatching. We will better define this within the introduction and methods.

- You may consider combining Figures 6 & 7 into one figure since they are discussed together. I found I kept looking at the wrong figure and combing them into one with clear labeling and different coloring may help.
  > Yes, fair enough. We will do this.

- Line 239: extra "An"
  > Deleted

- Line 344: that instead of which. This sentence is also oddly phrased and should be broken into two sentences.
  > Yes, this has been changed to:
  > *The most significant relationships with early settlement (Figure 10a) were; KE in the south (for two southern PI sites with some further importance at Lancelin and Alkimos), and both LC at 27º S during summer and early CC at 30º S for westward transport of phyllosoma (Jurien Bay, Lancelin (LC) and Port Gregory (CC)). These were in line with our predictions (Table 1).*

- Fig 11 legend. Delete the extraneous "model"
  > Deleted

- Line 412: extraneous period at the end of the sentence
  > Deleted

- Line 422: I assume 201109 should be 2009
  > Changed

- Line 470: "that" instead of "which"
  > Changed

---

## Referee Report (RR1)

GENERAL COMMENTS

Overall, I think the authors have done a good job responding to the previous comments with regards to improving overall clarity, particularly with regard to the introduction and clarifying eth project goals. Additionally, the authors have done a good job streamlining and improving the writing, and aligning the intro, methods, and R/D for added clarity, making the paper much easier to read and understand.

TECHNICAL COMMENTS

- Section 3.5, first sentence. This is an odd sentence structure. I actually think you can just delete this sentence. However, you say that seasonal and inter-annual variability were explored—how were they explored? Just graphically? This is not clear in the methods. Additionally, to make it congruent with the results, I would title this section "Exploration of variation in oceanographic conditions"
- Section 4. Instead of saying the results and discussion are laid out in 1, 2, 3, I would rather see a very brief summary of the general findings to start out the results/discussion section and then follow with the details in those three sections.
- There is an incomplete sentence in the figure legend for Figure 2 ("In particular for kinetic and eddy kinetic energy calculation."). I am not sure what this is supposed to say. There should also be a period after (b).
- The sentence in Len 233-234 is odd ("considering the large number of predictors…"). I would move this sentence to the start of the paragraph and rephrase it as something like "Table 1 shows each predictor variable the associated hypotheses tested." Then go to explain them. Additionally, the table 1 headings still says multiple regression analysis, but you changed it everywhere else to GAMs. Also, regarding the heading—I don't think the hypotheses are subsequent, but rather associated?
- Line 311: Water circulation "was" not "were"
- Line 542: delete "despite its exploratory nature"

---

## Author Response (AR2)

We want to thank this reviewer who has provided a detailed review of our manuscript and provided valuable feedback and suggestions, which have significantly helped improve our manuscript.
We have done our best to account for these comments and have responded individually (red text) to each of them below.

We hope that our response and modification to the manuscript will improve the manuscripts final publication.

Kind Regards

Jessica Kolbusz and co authors

GENERAL COMMENTS
**Overall, I think the authors have done a good job responding to the previous comments with regards to improving overall clarity, particularly with regard to the introduction and clarifying the project goals. Additionally, the authors have done a good job streamlining and improving the writing, and aligning the intro, methods, and R/D for added clarity, making the paper much easier to read and understand.**
Thank you for the supportive comment!

TECHNICAL COMMENTS
**Section 3.5, first sentence. This is an odd sentence structure. I actually think you can just delete this sentence. However, you say that seasonal and inter-annual variability were explored—how were they explored? Just graphically? This is not clear in the methods.**
We have deleted the first sentence. The first sentence of section 3.5 now reads: "Seasonal and inter-annual variability, not captured within the GAM, were explored graphically with the inclusion of moving means."
**Additionally, to make it congruent with the results, I would title this section "Exploration of variation in oceanographic conditions"**
Great suggestion. We have renamed this.

**Section 4. Instead of saying the results and discussion are laid out in 1, 2, 3, I would rather see a very brief summary of the general findings to start out the results/discussion section and then follow with the details in those three sections.**
Thank you for this comment. The start of Section 4 now reads:
"Our findings demonstrate that similar oceanographic conditions influence adjacent puerulus monitoring sites. The Leeuwin Current strength over the summer months has increased since the low puerulus settlement season, alongside a decrease in the Capes Current, suggesting a mismatch in the puerulus transport processes.  In addition, this period occurred alongside neutral ENSO conditions and cooler water over the likely *P. cygnus* pelagic distribution. These results and their discussion are in the following three sections: (1) time-series exploration of *P. cygnus* settlement between 2001 and 2017 and associated oceanographic conditions experienced by the larvae to represent each puerulus settlement season (May to April); (2) exploring the correlation of oceanographic conditions with settlement data through a general additive model analysis, and (3) inter-annual and seasonal oceanographic variability."

**There is an incomplete sentence in the figure legend for Figure 2 ("In particular for kinetic and eddy kinetic energy calculation."). I am not sure what this is supposed to say.**

This sentence has been removed.

**There should also be a period after (b).**

A period has been added.

**The sentence in Len 233-234 is odd ("considering the large number of predictors…"). I would move this sentence to the start of the paragraph and rephrase it as something like "Table 1 shows each predictor variable the associated hypotheses tested." Then go to explain them.**

Detail and rephrasing has been added here.

The paragraph now reads:

"Table 1 shows each predictor variable and the associated hypotheses for each annual value. The LC consistently flows southwards and is strongest over the winter months, possibly flooding the shelf. Therefore, over the winter months, this would likely positively affect late-stage phyllosoma successfully reaching the nearshore. Over the summer months, the LC strength, if stronger, would likely impede the survival of early-stage phyllosoma (Feng et al., 2011). Given the stronger opposing LC, the northward-flowing CC on the shelf would likely positively impact puerulus settlement (Muhling et al., 2008). Kinetic energy will likely positively impact the transportation of phyllosoma throughout their early pelagic life-cycle (Cetina-Heredia et al., 2019a; Hood et al., 2017). Similarly, cross-shelf transport offshore would likely increase the survival of phyllosoma and cross-shelf transport onshore after the pelagic phase would assist puerulus transportation onshore. *P. cygnus* spawning likely occurs sooner with an increased bottom water temperature, causing possible timing mismatches over the next 9 to 11 months (de Lestang et al., 2015). Conversely, warner water temperatures increase the rate of phyllosoma development, therefore likely increasing their survival (Phillips et al., 1978; de Lestang et al., 2015). If the IBSS were higher, there would be more spawning stock, therefore likely an increase in phyllosoma and eventual puerulus (de Lestang et al. 2012). These variables resulted in a total of 39 possible predictors of the late puerulus settlement (8 sites) and 33 possible predictors of the early puerulus settlement at sites (8) (Appendix A, Table A1). LC strength in summer and the late CC strength predictors for early settlement were omitted since they occur after early settlement each season. Given the predicted data availability and the spawning season is in the calendar year prior, the relationship between all predictors and puerulus settlement was limited to the 2001 to 2017 seasons."

**Additionally, the table 1 headings still says multiple regression analysis, but you changed it everywhere else to GAMs. Also, regarding the heading—I don't think the hypotheses are subsequent, but rather associated?**

The Table 1 heading has been changed to "Predictor variables and metrics used in the GAM analysis to investigate variability in puerulus settlement and associated hypothesis. The subscript *s* identifies the relativity of a month to the puerulus settlement season (May - Apr) in question. *s - 1* is within the season prior and *s + 1* is after. –ve denotes negative relationship and +ve denotes and positive relationship."

**Line 311: Water circulation "was" not "were"**

This has been changed

**Line 542: delete "despite its exploratory nature"**

We thank you for this comment. This has been deleted.